# Biophysical and physiological processes causing oxygen loss from coral reefs

**Cynthia B Silveira[1,2]\***, **Antoni Luque[2,3,4]**, **Ty NF Roach[5]**, **Helena Villela[6]**, **Adam Barno[6]**, **Kevin Green[1]**, **Brandon Reyes[1]**, **Esther Rubio-Portillo[7]**, **Tram Le[1]**, **Spencer Mead[1]**, **Mark Hatay[1,2]**, **Mark JA Vermeij[8,9]**, **Yuichiro Takeshita[10]**, **Andreas Haas[11]**, **Barbara Bailey[4]**, **Forest Rohwer[1,2]**

[1]Department of Biology, San Diego State University, San Diego, United States; [2]Viral Information Institute, San Diego State University, San Diego, United States; [3]Computational Science Research Center, San Diego State University, San Diego, United States; [4]Department of Mathematics and Statistics, San Diego State University, San Diego, United States; [5]Hawaii Institute of Marine Biology, University of Hawaii at Mānoa, Kāneohe, United States; [6]Department of Microbiology, Rio de Janeiro Federal University, Rio de Janeiro, Brazil; [7]Department of Physiology, Genetics and Microbiology, University of Alicante, Alicante, Spain; [8]CARMABI Foundation, Willemstad, Curaçao; [9]Department of Freshwater and Marine Ecology, Institute for Biodiversity andEcosystem Dynamics, University of Amsterdam, Amsterdam, Netherlands; [10]Monterey Bay Aquarium Research Institute, Moss Landing, United States; [11]NIOZ Royal Netherlands Institute for Sea Research, Utrecht University, Texel, Netherlands

**\*For correspondence:**
cynthiabsilveira@gmail.com

**Competing interests:** The authors declare that no competing interests exist.

**Reviewing editor:** Evan Howard,

**Abstract** The microbialization of coral reefs predicts that microbial oxygen consumption will cause reef deoxygenation. Here we tested this hypothesis by analyzing reef microbial and primary producer oxygen metabolisms. Metagenomic data and in vitro incubations of bacteria with primary producer exudates showed that fleshy algae stimulate incomplete carbon oxidation metabolisms in heterotrophic bacteria. These metabolisms lead to increased cell sizes and abundances, resulting in bacteria consuming 10 times more oxygen than in coral incubations. Experiments probing the dissolved and gaseous oxygen with primary producers and bacteria together indicated the loss of oxygen through ebullition caused by heterogenous nucleation on algae surfaces. A model incorporating experimental production and loss rates predicted that microbes and ebullition can cause the loss of up to 67% of gross benthic oxygen production. This study indicates that microbial respiration and ebullition are increasingly relevant to reef deoxygenation as reefs become dominated by fleshy algae.

## Introduction

Anthropogenic pressures are shifting the composition of reef primary producer communities from calcifying to non-calcifying organisms globally (*Cinner et al., 2016*; *Smith et al., 2016*). These changes occur as fleshy algae gain competitive advantage over corals through their interaction with heterotrophic microbes (*Barott and Rohwer, 2012*; *Jorissen et al., 2016*). High Dissolved Organic Carbon (DOC) release rates by fleshy algae stimulate overgrowth of heterotrophic bacteria (*Haas et al., 2011*; *Kelly et al., 2014*; *Nelson et al., 2013*). The exacerbated heterotrophic growth creates hypoxic zones at the coral-algae interface that kill corals (*Haas et al., 2013a*; *Haas et al., 2014b*; *Roach et al., 2017*; *Smith et al., 2006*). At the reef-scale, algae dominance stimulates heterotrophic bacteria in the water above the reef and leads to an increase in microbial biomass and

**eLife digest** Rising water temperatures, pollution and other factors are increasingly threatening corals and the entire reef ecosystems they build. The potential for corals to resist and recover from the stress these factors cause ultimately depends on their ability to compete against fast-growing fleshy algae that can rapidly take over the reefs.

Living on the fleshy algae, the coral and in the surrounding water are communities of bacteria and other microbes that help maintain the health of the coral reef. Both corals and algae modify the chemical and physical environment of the reef to alter the composition of the microbial communities for their own benefit. Algae, for instance, release large amounts of sugars and other molecules of organic carbon into the water. These carbon molecules are then taken up by the bacteria, along with oxygen, to produce chemical energy via a process called respiration. This could cause the levels of oxygen in the water to decrease, potentially damaging the corals and creating more open space for the algae.

Previous studies have revealed how communities of microbes on coral reefs use organic carbon, but it remains unclear how they affect the levels of oxygen in the reefs. To address this question, Silveira et al. used an approach called metagenomics to analyze the bacteria in samples of water from 87 reefs across the Pacific and the Caribbean, and also performed experiments with reef bacteria grown in the laboratory.

The experiments showed that bacteria growing in the presence of fleshy algae became larger and more abundant than bacteria growing near corals, resulting in the water containing lower levels of oxygen. Furthermore, the fleshy algae produced bubbles of oxygen that were released from the water. Silveira et al. developed a mathematical model that predicted that these bubbles, combined with the respiration of bacteria that live near algae, caused the loss of 67% of the oxygen in the water surrounding the reef.

These findings represent a fundamental step towards understanding how changes in the levels of oxygen in water affect the ability of coral reefs to resist and recover from stress.

energetic demands, described as reef microbialization (*Haas et al., 2016*; *McDole et al., 2012*; *Silveira et al., 2015*; *Silveira et al., 2017*). Overgrowth of heterotrophic bacteria creates a feedback loop of coral death, opening space for more algae overgrowth and microbial biomass accumulation (*Dinsdale and Rohwer, 2011*; *Rohwer et al., 2002*). While DOC-microbe relationships have been extensively described both experimentally and in situ, the oxygen fluxes within the microbialization feedback loop are not fully understood.

Differences in microbial responses to coral and algae dominance stem from the physiology of these primary producers. Calcifying organisms, including scleractinian corals and crustose coralline algae (CCA), invest 50% to 80% of their daily fixed carbon in respiration to sustain the energetic costs of calcification (*Hatcher, 1988*; *Houlbrèque and Ferrier-Pagès, 2009*; *Tremblay et al., 2012*). Fleshy macroalgae allocate 10% to 30% to respiration, and release up to 60% of their primary production as dissolved organic carbon (DOC) in the water (*Cheshire et al., 1996*; *Crossland, 1987*; *Jokiel and Morrissey, 1986*; *Peninsula et al., 2007*). Fleshy algae also allocate a higher proportion of their daily synthesized carbon to biomass compared to corals, sustaining high herbivory pressure (*Duarte and Cebrián, 1996*; *Falkowski et al., 1984*; *Tremblay et al., 2016*). Together, these processes are predicted to increase DOC-to-$O_2$ ratios in coral exudates compared to algae, but experimental data have shown the opposite to be true: the ratio between DOC and $O_2$ released by fleshy algae in bottle incubations is higher compared to corals (*Haas et al., 2013b*; *Haas et al., 2011*).

Heterotrophic microbes growing on algal exudates produce fewer cells per unit of carbon consumed compared to growth on coral exudates (*Haas et al., 2011*; *Nelson et al., 2013*; *Silveira et al., 2015*). Microbes growing on algae-dominated reefs shift their metabolism from classic glycolysis (Embden–Meyerhof–Parnas pathway, EMP) towards Pentose Phosphate (PP) and Entner-Doudoroff (ED) pathways (*Haas et al., 2016*). All three pathways consume glucose in a series of redox reactions that produce pyruvate, however they transfer electrons to distinct reduced coenzymes that act as intermediary electron transfer molecules (*Klingner et al., 2015*; *Spaans et al., 2015*). These reduced coenzymes have different fates in the cell, and as a result, the classic glycolic

pathway generates more ATP and consumes more oxygen, while ED and PP leave a higher fraction of electrons accumulated as biomass (*Fuhrer and Sauer, 2009*). These differences predict that microbes found in microbialized reefs have different oxygen consumption rates when compared to healthy reefs (*Flamholz et al., 2013*; *Stettner and Segrè, 2013*).

Fleshy turf and macroalgae release up to three times more $O_2$ in the surrounding seawater than calcifying organisms (*Haas et al., 2011*; *Naumann et al., 2010*; *Nelson et al., 2013*; *Silveira et al., 2015*). Yet, algae-dominated systems have consistently lower $O_2$ standing stocks (*Calhoun et al., 2017*; *Haas et al., 2013b*; *Martinez et al., 2012*). Fleshy algae can produce $O_2$ bubbles through heterogeneous nucleation resulting from $O_2$ super-saturation at the algae's surface (*Kraines et al., 1996*). The gaseous $O_2$ in bubbles is not detected by dissolved $O_2$ instruments, and may cause underestimation of autotrophy in oxygen-based primary productivity methods (*Atkinson and Grigg, 1984*; *Kraines et al., 1996*). While several studies recognize bubble formation on algal surfaces in benthic ecosystems, the fraction of photosynthetic oxygen lost as bubbles from coral reef primary producers has never been quantified (*Freeman et al., 2018*; *Odum and Odum, 1955*).

Here we test the hypothesis that $O_2$ loss in coral reefs is a result of the high microbial oxygen consumption and the biophysical loss through ebullition from algae surfaces. The analysis of organic carbon consumption pathways encoded in bacterial metagenomes from 87 reefs in the Atlantic and Pacific combined with experimental quantification of cell-specific DOC and oxygen consumption showed that fleshy macroalgae and microbes remove larger amounts of photosynthetic $O_2$ by ebullition and respiration compared to corals. As a result, the incubation bottles have similar to lower oxygen concentrations, providing a mechanistic explanation for the lower dissolved $O_2$ observed on reef ecosystems shifted to algal dominance.

## Results

### Abundance of carbon metabolism genes across algal cover gradient

Different carbon consumption pathways employed by bacteria are associated with varying levels of oxygen consumption rates (*Russell and Cook, 1995*). To investigate the carbon metabolic pathways selected among reef microbes, 87 metagenomes from reefs across a gradient in algal cover were analyzed for the relative abundance of genes encoding rate-limiting enzymes of central carbon metabolism (*Figure 1—source data 1*). On coral reefs, metagenomic data reflect the strain-level genomic adaptation that occurs within hours, the timescale of residence time depending on the tides and current regime (*Nelson et al., 2011*). This selection occurs as offshore microbial communities are transported onto the reef environment and water masses get enriched with benthic exudates changing their biogeochemistry (*Nelson et al., 2011*; *Kelly et al., 2014*). In our dataset, microbial biomass in the reef boundary layer increased with fleshy algae cover (linear regression, p=6.25 x 10$^{-8}$, $R^2$ = 0.29).

A dimension-reduction random forest regression analysis was applied to genes encoding rate-limiting enzymes of central carbon metabolism, stress response and control genes, for a total of 23 variables (*Supplementary file 1*), using fleshy algae cover as predicted variable. Random forest is a non-parametric method that does not rely on normality, and along with other machine-learning approaches, is the method of choice for the identification of a robust subset of variables within complex meta-omics data (*Li, 2015*; *Liu et al., 2019*; *Thompson et al., 2019*). The relative abundance of these genes could explain 14.3% of the variance in fleshy algae cover. The genes with highest and significant explanatory power in the permutation test were phosphoenolpyruvate carboxylase, aspartate aminotransferase, oxoglutarate dehydrogenase, glucose 6-phosphate dehydrogenase and KDPG aldolase (*Figure 1* displays the enzyme genes with highest explanatory power according to their increasing mean squared error: 53.54%, 22.03%, 23.99%, 26.08% and 17.89%, and permutation test p-values = 0.009, 0.01, 0.03, 0.01 and 0.03, respectively; *Figure 1—source data 1*).

Phosphoenolpyruvate carboxylase and aspartate aminotransferase are involved in anaplerotic routes, reactions that replenish the Krebs cycle when its intermediaries are diverted for biosynthesis. Glucose 6-phosphate dehydrogenase shunts glucose to both Entner-Doudoroff and Pentose Phosphate pathways, and KDPG aldolase is unique to the Entner-Doudoroff pathway. These four enzyme genes had a positive relationship with fleshy algae cover (identified by random forest by their increasing mean squared error, followed by permutation test with p-values p=0.009, 0.01,

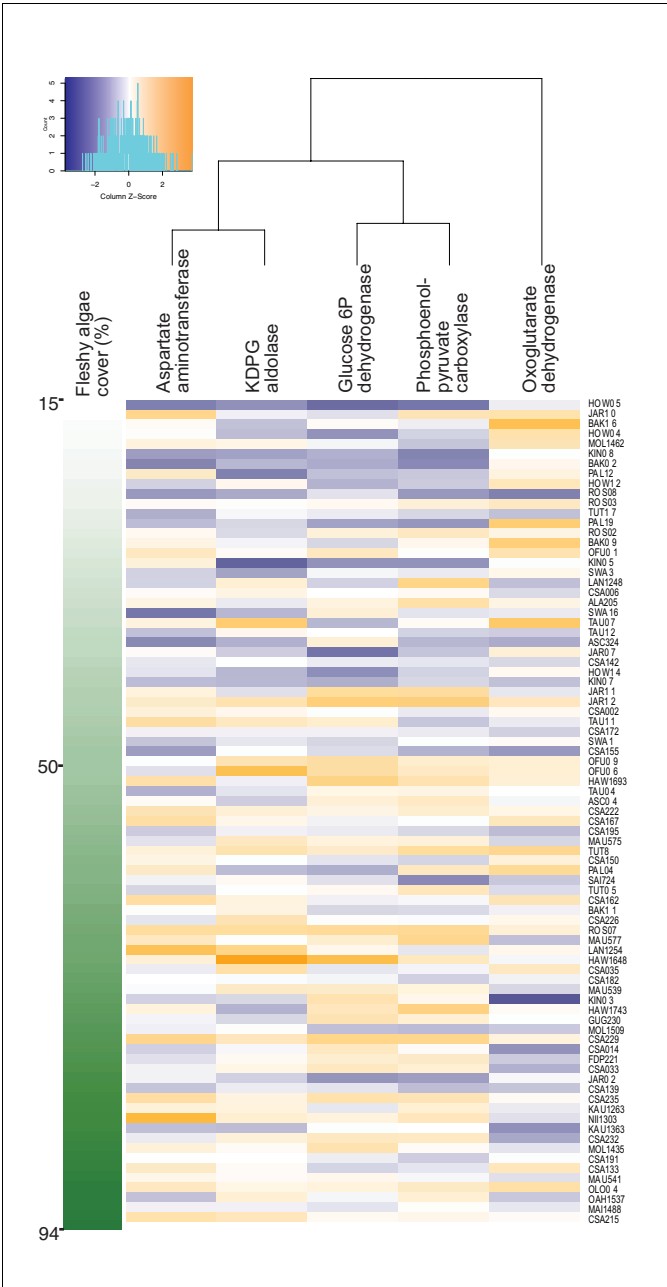

**Figure 1.** Increase in genes encoding anabolic pathways and decrease in genes encoding carbon oxidation reactions with increasing fleshy algae cover in situ. Relative abundances of genes encoding rate-limiting enzymes in reef metagenomes are plotted in relation to fleshy algae cover (sum of fleshy turf and macroalgae). Individual metagenomes are listed as rows and sorted by the fleshy algae cover, ranging from 15% to 94.1%. Enzyme gene abundance was scaled by column to allow between-enzyme comparisons as indicated by the z-score, where blue indicates low relative abundance and orange indicates high relative abundance. Only enzyme genes significantly predicting fleshy algae cover in the random forest analysis are shown. A complete list of enzymes and abundances is provided as *Figure 1—source data 1*.

The online version of this article includes the following source data and figure supplement(s) for figure 1:

**Source data 1.** Geographic location, microbial biomass, benthic cover and metagenomic data of 87 reef sites in the Pacific and Caribbean.

**Figure supplement 1.** Diagnostic plot of the mean squared error in the permutational regression random forest analysis of enzyme gene relative abundances using percent fleshy algae cover as response variable.

**Figure supplement 2.** Most abundant species in the microbial metagenomes.

*Figure 1 continued on next page*

*Figure 1 continued*

**Figure supplement 3.** Diagnostic plot of the mean squared error in the permutational regression random forest analysis of bacterial species using percent fleshy algae cover as response variable.

0.01 and 0.03, respectively), and participate in pathways of incomplete carbon oxidation, that is, lower oxygen consumption. Oxoglutarate dehydrogenase relative abundance had an inverse relationship with fleshy algae cover (random forest increasing mean squared error, p=0.03). This enzyme catalyzes an oxidative decarboxylation step in the Krebs cycle, a pathway that produces most of the NADH that fuels oxygen consumption by oxidative phosphorylation. These results suggest that the bacterial community is being increasingly selected for pathways related to overflow metabolism and incomplete carbon oxidation at high algal cover, which would predict a decoupling between organic carbon and oxygen consumption rates by microbes.

The gene *rpoB* (RNA polymerase) was used as a control and did not vary in relative abundance across the algal cover gradient (*Figure 1—source data 1*). Genes involved in oxidative stress did not show significant relationships either. To test if the changes in enzyme gene relative abundances could be explained by taxonomic shifts of bacterial genera that consistently display larger genome sizes and duplications in the genes described above, the relationship between fleshy algae cover and taxonomic profiles at the genus and species levels was tested using random forest. The relative abundances of each taxon were calculated accounting for genome size (*Cobián Güemes et al., 2016*). Seven species significantly changed in abundance across the algae cover gradient (*Figure 1—source data 1*, *Prochlorococcus marinus*, *Fibrisoma limi*, *Prolixibacter bellariivorans*, *Pelagibacter ubique*, *Ruegeria mobilis*, *Phaeobacter italicus*, and *Psychrobacter pacificensis*). At the genus level, five taxa (*Prochlorococcus*, *Thermotoga*, *Methanobacterium*, *Leuconostoc*, and *Rhodopirellula*) changed their abundance with increasing fleshy algae cover. We searched if the taxa varying in abundance consistently display copy number variations in the genes that significantly vary with algae cover. The only taxon that could potentially explain the abundance trends in enzyme genes was *Prochlorococcus*, which lacks an oxoglutarate dehydrogenase gene but encodes all the other four enzymes. However, there was no significant relationship between the abundances of *Prochlorococcus* and oxogluratae dehydrogenase (p=0.26 in the linear regression of oxoglutarate dehydrogenase and $log_{10}$-transformed *Prochlorococcus* abundance). *Figure 1—source data 1* shows the relative abundances of the 20 most abundant species that together made up 97.8 ± 3.0% of the identified reads at species level in the metagenomes.

## Bacterial DOC and $O_2$ consumption

To test the hypothesis of a decoupling between organic carbon and oxygen consumption by heterotrophic bacteria predicted by the metagenomic analysis, cell-specific carbon and $O_2$ consumption rates were obtained by incubating reef heterotrophic bacteria with coral and algae exudates (*Figure 2—source data 1*; Experiment one in Materials and methods). The amount of DOC consumed per cell was higher, that is, low cell yields per carbon consumed, in fleshy macroalgae treatments compared to coral and control treatments (*Figure 2A*, Kruskal-Wallis $\chi^2(2)=13.7$, p=0.001; Wilcoxon, p=0.005 and 0.008 for pairwise comparisons of algae vs controls and corals, respectively). However, the amount of $O_2$ consumed per cell was not different between treatments (*Figure 2B*, Kruskal-Wallis $\chi^2(2)=0.5$, p=0.77).

## Bacterial abundances and cell volumes

Incomplete carbon oxidation and low oxygen consumption relative to organic carbon are hallmarks of increases in overflow metabolism and organic carbon accumulation as biomass. To test whether low $O_2$-to-DOC consumption ratios in algae exudates are causing biomass accumulation, heterotrophic bacteria were incubated in coral, CCA and fleshy turf and macroalgae exudates and their biomass was quantified accounting for changes in both abundance and cell volume. The change in bacterial abundance in all primary producer bottles was higher compared to controls (*Figure 3A*, Kruskal-Wallis $\chi^2(4)=20.6$, p=0.0003, Wilcoxon p=0.04, 0.02. 0.02. and 0.02 for pairwise comparisons of coral, CCA, turf and macroalgae against control, *Figure 3—source data 1*). The change in abundance was higher in fleshy algae treatments compared to calcifying treatments (Kruskal-Wallis $\chi^2(2)$

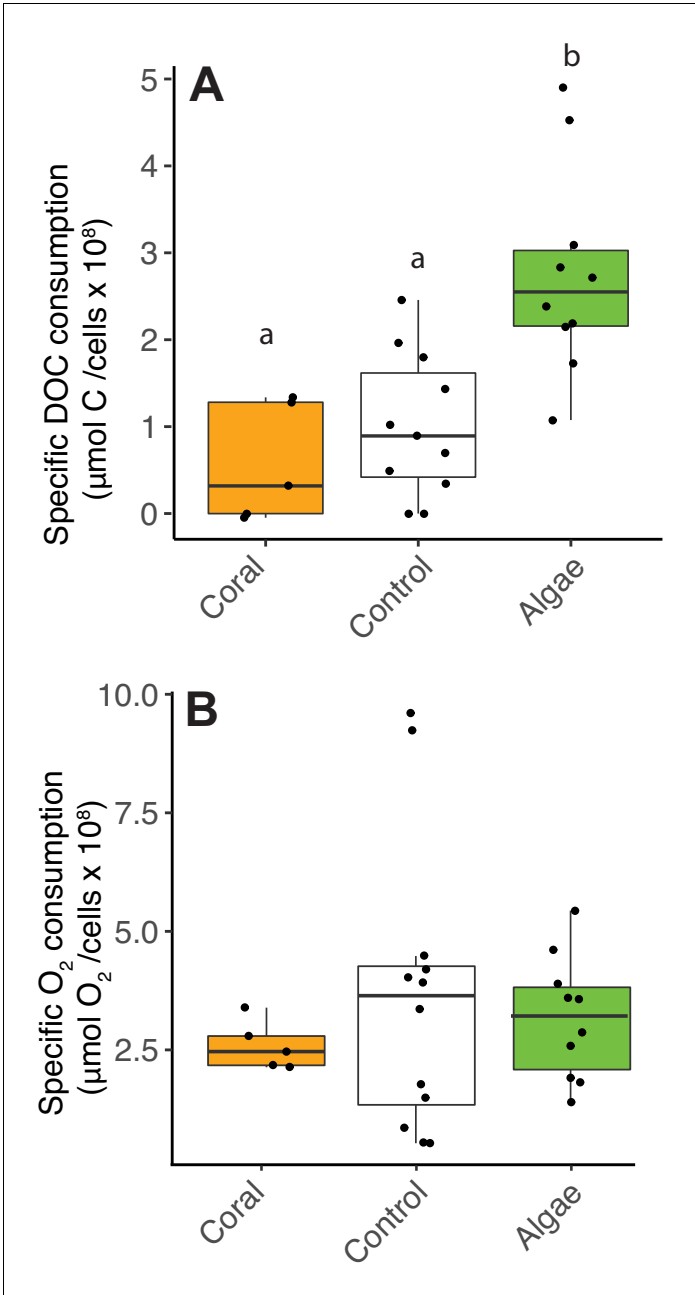

**Figure 2.** Decoupling between microbial DOC and $O_2$ consumption in fleshy macroalgae exudates. Cell-specific carbon and $O_2$ consumption data from Experiment 4: dark incubations of microbial communities in primary producer exudates. (A) Cell-specific DOC consumption. (B) Cell-specific $O_2$ consumption. Primary producer treatments had a significant effect on specific DOC consumption only (Kruskal-Wallis p<0.05) and letters above boxes indicate p<0.05 for Wilcoxon pairwise tests with FDR correction. Orange indicates calcifying and green indicates fleshy organisms. Boxes depict the median and the first and third quartiles. Whiskers extend to 1.5 times the size of the interquartile ranges. Outliers are represented as dots above or below whiskers.

The online version of this article includes the following source data for figure 2:

**Source data 1.** Changes in DOC, $O_2$ and cell abundances in experimental bottles.

=19.1, p=6.9×$10^{-5}$, Wilcoxon p=0.001 for pairwise comparison of fleshy vs calcifying). Microbial cell volume changed differently in treatments during incubations (*Figure 3B*, Kruskal-Wallis $\chi^2$(5) =1235.2, p<2×$10^{-16}$, *Figure 3—source data 2*). Bacteria growing in control bottles showed no

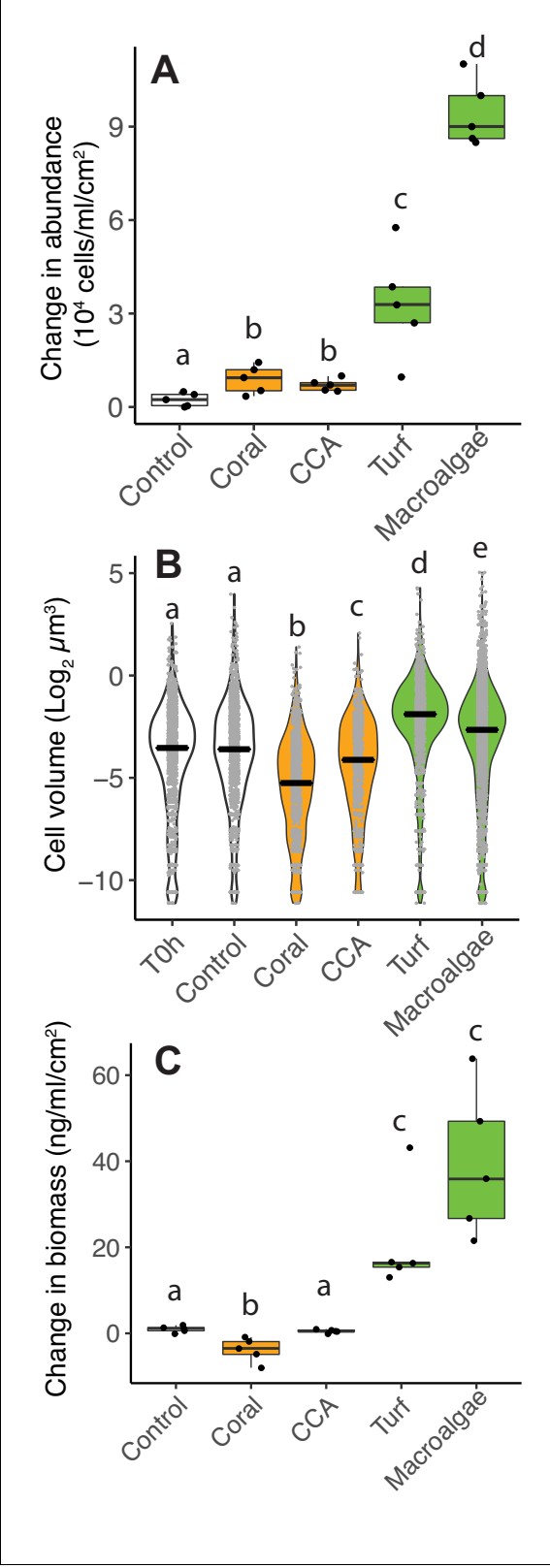

**Figure 3.** Differences in microbial biomass accumulation in primary producer exudates. (A) Changes in microbial abundance normalized by primary producer surface area. (B) Cell volume distributions. (C) Changes in total microbial biomass, accounting for abundance and cell volume, normalized by primary producer surface area. Primary producer treatments had a significant effect on all three microbial variables (Kruskal-Wallis p<0.05), and

*Figure 3 continued on next page*

*Figure 3 continued*

letters above boxes indicate p<0.05 for Wilcoxon pairwise tests with FDR correction. Orange indicates calcifying and green indicates fleshy organisms. Boxes depict the median and the first and third quartiles. Whiskers extend to 1.5 times the size of the interquartile ranges. Outliers are represented as dots above or below whiskers.

The online version of this article includes the following source data and figure supplement(s) for figure 3:

**Source data 1.** Cell abundances and biomass in experimental bottles.

**Source data 2.** Microbial cell volumes in experimental bottles.

**Figure supplement 1.** Changes in microbial abundance (**A**), cell volume (**B**) and total biomass (**C**) stimulated by coral and algal exudates.

change in volume (Wilcoxon, p=0.56 for pairwise comparison of control vs T0h). Cell volume decreased when bacteria grew on coral and CCA exudates, but increased when growing on turf and macroalgae exudates, and showed no change in controls (Wilcoxon $p<2\times10^{-16}$ for all pairwise comparisons vs T0h and vs control).

When taking abundance and cell volume into account, the change in total bacterial biomass was significantly different among treatments (*Figure 3C*, Kruskal-Wallis $\chi^2(4)=21.6$, p=0.0002, *Figure 3—source data 1*). Coral exudates decreased total bacterial biomass, while turf and macroalgae increased total bacterial biomass, and CCA lead to no change (Wilcoxon, p<0.05 for pairwise comparisons of coral, turf and macroalgae vs control). The change in biomass in the two fleshy algae together was greater than in the calcifying organisms together (Kruskal-Wallis $\chi^2(2)=19.1$, $p=7.1\times10^{-5}$; Wilcoxon, p=0.0005 for pairwise comparison of fleshy vs calcifying). This same pattern was observed in two independent experiments, one performed in the island of Curaçao, in the Caribbean (*Figure 3*) and one performed at the Hawaiian Institute of Marine Biology (HIMB, *Figure 3—figure supplement 1*), with distinct sets of primary producers, suggesting that this result may be broadly applicable to other reefs (*Figure 3—source datas 1* and *2*).

## Oxygen production, consumption, and ebullition

While bacteria became larger and more abundant when growing on algae exudates, their oxygen consumption was lower than that predicted by their DOC consumption: the theoretical value for full carbon oxidation through aerobic metabolism is 1:1, changing due overflow metabolism, futile cycles, uncoupling, and other processes reviewed in *Russell and Cook (1995)*. To resolve the oxygen budget, heterotrophic bacteria were incubated together with the primary producers in POP (Peripheral Oxygen Production) incubation chambers that allow to quantify $O_2$ in both dissolved and gas fractions. Net dissolved $O_2$ production was significantly different among primary producer treatments (*Figure 4A*, Kruskal-Wallis $\chi^2(4)=19.5$, p=0.0006, *Figure 4—source data 1*). Primary producers showed higher net dissolved $O_2$ production compared to controls (Wilcoxon with FDR-corrected pairwise comparisons p=0.004, 0.01, 0.01, 0.004 for comparisons between controls vs coral, CCA, turf and macroalgae, respectively). Net dissolved $O_2$ production was lower in macroalgae incubations, but not significantly different between fleshy and calcifying organisms (5.99 ± 2.29 and 6.33 ± 2.20 µmol $cm^{-2}$ $day^{-1}$ for fleshy and calcifying organisms, respectively, mean ± SE; Wilcoxon p=0.68). There was a significant difference between gaseous $O_2$ production observed in most primary producer bottles, while no gas was observed in control bottles (*Figure 4B*, Kruskal-Wallis $\chi^2(4)=30.2$, p=4.361e-06, Wilcoxon p=0.03, 0.002, 0.002, and 0.002 for pairwise comparisons between controls vs corals, CCA, turf and macroalgae, respectively). Fleshy organisms produced significantly more gaseous $O_2$ than calcifying organisms (0.42 ± 0.35 and 2.33 ± 1.35 µmol $cm^{-2}$ $day^{-1}$ for calcifying and fleshy, respectively, mean ± SE; Wilcoxon p=0.0001 for comparisons between fleshy vs controls and vs coral). Fleshy macroalgae had the highest fraction of net photosynthetic $O_2$ (sum of dissolved and gaseous $O_2$) released in the form of gas (37.33 ± 8.34%), followed by turf algae (13.78 ± 1.33%), CCA (10.19 ± 2.88%), and corals (5.00 ± 5.55%) (*Figure 4C*). The difference in the fraction of $O_2$ accumulated as gas was significant among treatments (Kruskal-Wallis $\chi^2(4)=29.2$, p=7.067e-06), being higher in fleshy organisms (Kruskal-Wallis $\chi^2(2)=26.3$, p=1.904e-06; Wilcoxon p<9.5e-05 for fleshy vs calcifying).

In the POP experiments, the oxygen in the gas phase was close to equilibrium with the dissolved oxygen, and in some of the fleshy algae and CCA, the dissolved oxygen concentration was above

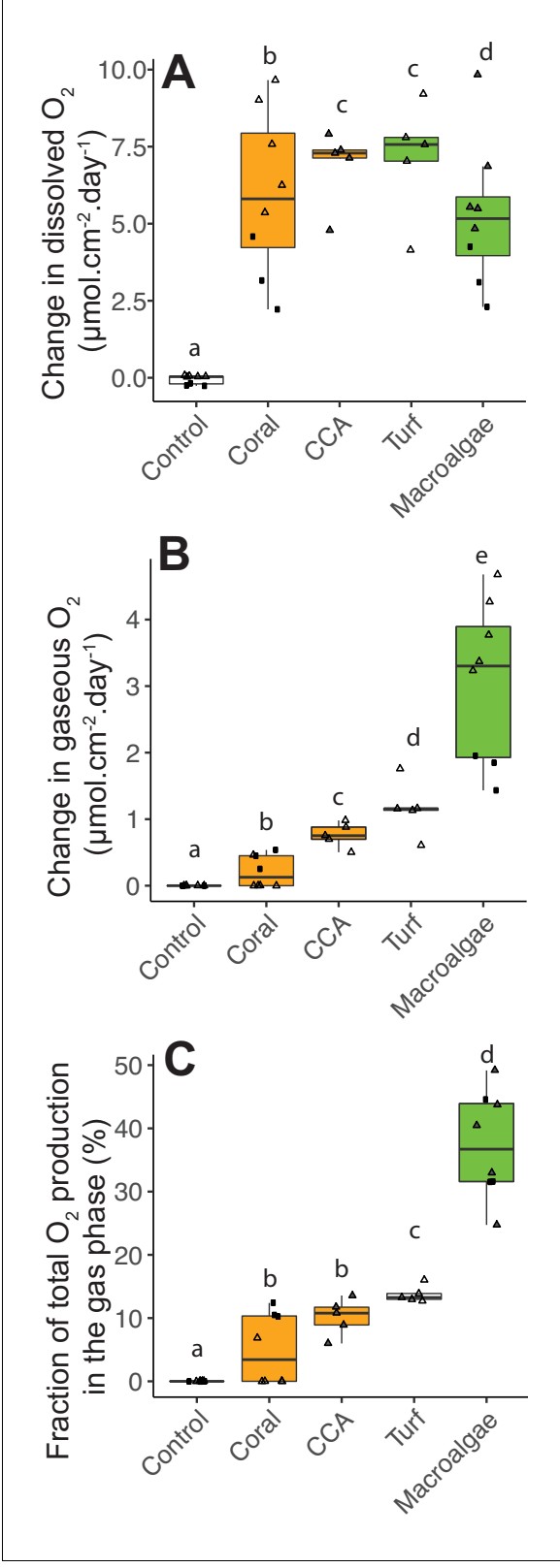

**Figure 4.** Photosynthetic $O_2$ loss as gas from fleshy macroalgae. (**A**) Dissolved $O_2$ production rates normalized by organism surface area. (**B**) Gaseous $O_2$ production rates normalized by organism surface area. (**C**) Fraction of total photosynthetic $O_2$ production in the form of gas. Solid squares correspond to Experiment 1 and triangles to Experiment 2. In A, the gray-filled triangles indicate replicates where dissolved oxygen was above the accuracy

*Figure 4 continued on next page*

*Figure 4 continued*

range of the probe used, and lower bound values (within probe accuracy range) were utilized. In C, the gray-filled triangles indicate the estimated values associated with lower bound values from A. Primary producer had a significant effect on all three variables (Kruskal-Wallis p<0.05), and letters above boxes indicate p<0.05 for Wilcoxon pairwise tests with FDR correction. Orange indicates calcifying and green indicates fleshy organisms. Boxes depict the median and the first and third quartiles. Whiskers extend to 1.5 times the size of the interquartile ranges. Outliers are represented as dots above or below whiskers.

The online version of this article includes the following source data for figure 4:

**Source data 1.** $O_2$ concentrations in dissolved and gas phases in POP incubation bottles.

accuracy limits of the probe. During the course of the experiment, we observed that a headspace was formed by bubbles that were formed on the algae surfaces and rose to the top of the incubation bottles. To quantify ebullition rates and to test whether this effect would be observed in an open system, bubble production was quantified in open-tank experiments using image analysis (*Figure 5A—source data 1*). The green fleshy algae *Chaetomorpha* had the highest bubble production rates (10.3 ± 0.45 bubbles per min per $dm^2$, mean ± SE; *Video 1*), followed by the fleshy red algae *Gracilaria* sp. (1.29 ± 1.21 bubbles per min per $dm^2$). Bubble production by the two coral species analyzed was close to zero (0.1 ± 0.14 and 0.006 ± 0.02 bubbles per min per $dm^2$ for *Favia* sp. and *Montipora* sp., respectively). Ebullition rate was significantly higher in fleshy compared to calcifying organisms (Kruskal-Wallis $\chi^2(3)$=349.46, p =<2.2e-16; Wilcoxon p<2e-16 for fleshy vs calcifying). Bubbles were observed forming on the algal surfaces, detaching and rising to the atmosphere starting as early as 20 min into the experiment. The ebullition rate in the open-tank experiment was within the range of ebullition rates estimated to be necessary to explain the volume and oxygen concentrations observed in the POP experiments (estimates indicated by the gray bar in *Figure 5A*). The dissolved oxygen concentration also increased through the course of the incubations (*Figure 5B*), however, there was no relationship between the rate of bubble production and the dissolved oxygen concentration (linear regression p=0.16, $R^2$ = 0.04) or the change in oxygen concentration over time (linear regression p=0.27, $R^2$ = 0.03).

The mean bubble diameter produced by the algae *Gracialaria* sp. was larger than that produced by the algae *Chaetomorpha* sp. (*Figure 5C* and *Figure 5—figure supplement 1*; 0.79 ± 27 mm in diameter and 0.64 ± 0.23 mm for *Gracialaria* sp. and *Chaetomorpha* sp., respectively, however these differences were not statistically significant). The incubation of these same algae species at higher PAR values (300 to 650 μmol quanta $m^{-2}$ $s^{-1}$, normally observed on a reef in a sunny day) yielded different bubble production rates: at 300 μmol quanta $m^{-2}$ $s^{-1}$, the algae *Chaetomorpha* sp. had the highest ebullition rates (*Figure 5C* and *Video 2*, 16.2 ± 1.9 bubbles per min per $dm^2$). The rate decreased to 0.9 ± 0.04 at 650 μmol quanta $m^{-2}$ $s^{-1}$. The algae *Gracilaria* sp. also produced fewer bubbles per minute at 650 μmol quanta $m^{-2}$ $s^{-1}$ (0.9 ± 0.08).

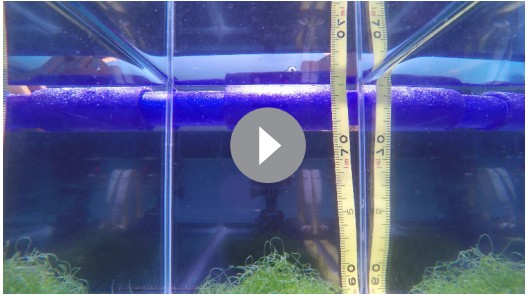

**Video 1.** Representative video of open-tank experiment with *Chaetomorpha* sp. at 150 umol quanta $m^{-2}$ $s^{-1}$. This specific video fragment was recorded 20 min after the beginning of the experiment.
https://elifesciences.org/articles/49114#video1

## Prediction of oxygen loss by microbial respiration and ebullition

To predict the proportion of the benthic gross oxygen production that is lost through microbial respiration and ebullition, a weighted linear model combined the cell-specific respiration rates from *Figure 2*, the microbial abundances sustained by each primary producer from POP experiments, and the rates of oxygen ebullition from *Figure 5*. Each of these rates was applied to an idealized reef of 1 $m^2$ benthic surface covered by fleshy green algae and/or scleractinian coral, and the microbial community within 1 $m^3$ of the water column above. Both the microbial respiration and ebullition rates increased with increasing algal cover in this idealized model (*Figure 5D—source data 2*). At 0% algal cover,

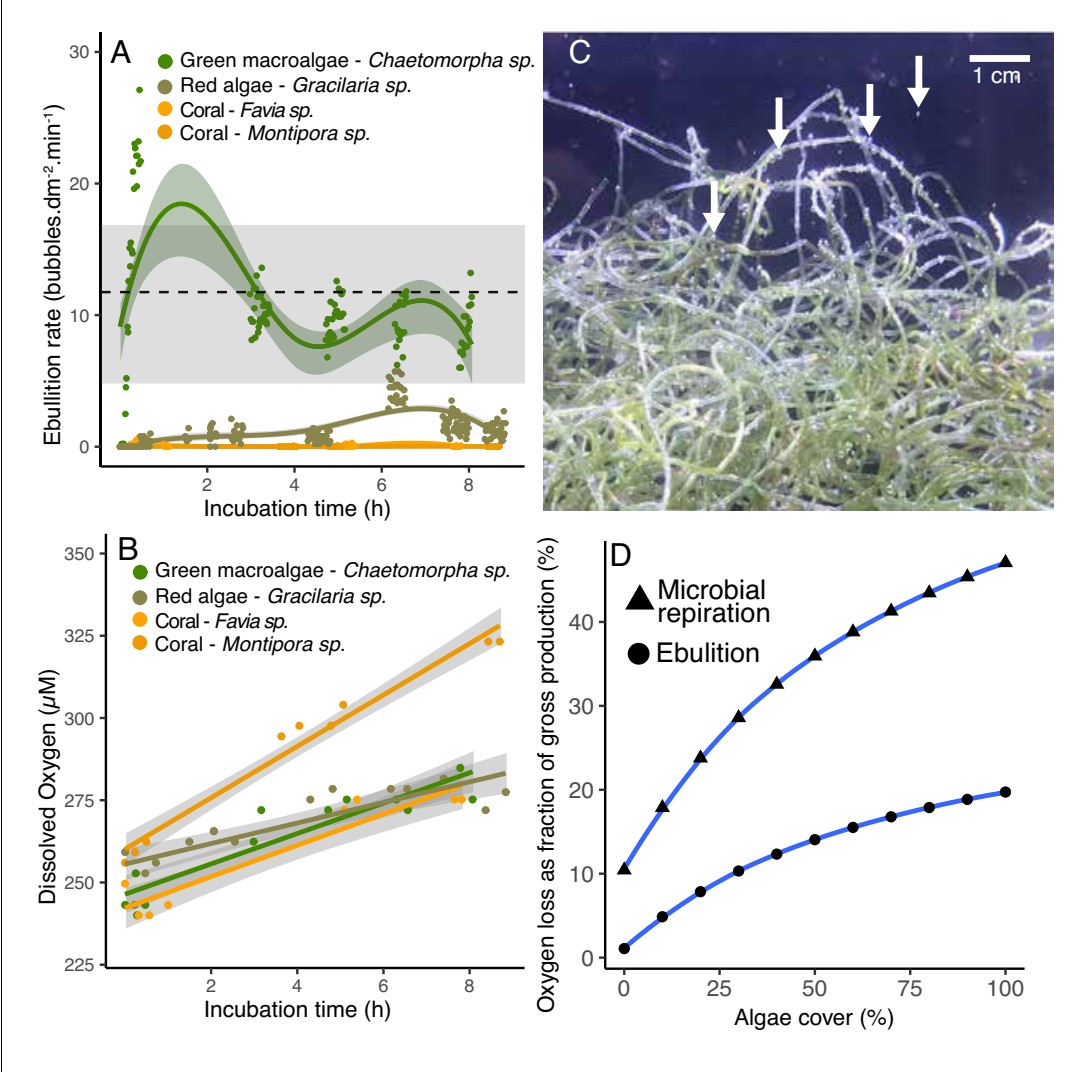

**Figure 5.** Ebullition by fleshy algae. (**A**) Rate of bubble production normalized by organism surface area in open-tank experiments. The green symbols indicate fleshy algae and the orange ones indicate scleractinian corals. The dotted line indicates the mean and the gray bar indicates the range of ebullition rates estimated to explain the amount of gaseous oxygen observed in POP experiment 2 (**Figure 4**). (**B**) Dissolved oxygen concentrations over time in the same open-tank experiments depicted in 5A. (**C**) Representative image of bubbles formed on the surface of the green algae *Chaetomorpha* sp. incubated in an open tank at 300 PAR. (**D**) Predicted loss of oxygen through microbial respiration and ebullition as a percentage of the gross benthic oxygen production in a model reef with varying algae and coral cover (x axis). Oxygen loss was estimated by a weighted linear model incorporating per-cell respiration rates shown in **Figure 2**, microbial abundances sustained by primary producers shown in **Figure 3**, and ebullition rates shown in **Figure 5A**.

The online version of this article includes the following source data and figure supplement(s) for figure 5:

**Source data 1.** Ebullition rates and dissolved O$_2$ in open tank experiments.
**Source data 2.** Model predictions of ebullition and microbial respiration contributions to O$_2$ loss.
**Figure supplement 1.** Bubble size distribution in open-tank experiments.

10.4% of the benthic oxygen production is consumed by microbes, compared to 35.9% at 50% algal cover, and 47.0% at 100% algal cover. The amount of oxygen loss by ebullition is predicted to increase from 1.0% at 0% algal cover to 14.0% at 50% algal cover and 19.7% at 100% algae cover. Combined, microbial respiration and ebullition are predicted to consume 50% of the gross oxygen production at 50% algal cover, and 66.7% of the oxygen production at 100% algal cover.

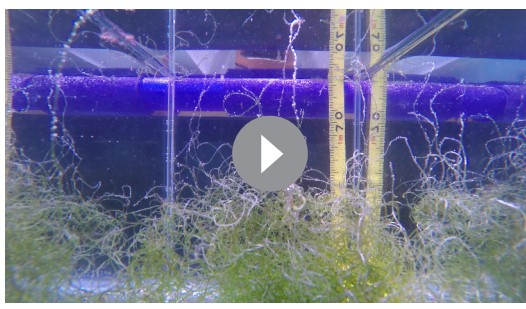

**Video 2.** Representative video of open-tank experiment with *Chaetomorpha* sp. at 300 μmol quanta m$^{-2}$ s$^{-1}$. This specific video fragment was recorded 7 hr after the beginning of the experiment.
https://elifesciences.org/articles/49114#video2

## Discussion

Due to high per cell activity and relative size, a small change in bacterial biomass may represent large shifts in ecosystems energy allocation, biogeochemistry landscape, and trophic relationships (*DeLong et al., 2010*; *McDole et al., 2012*). In coral reefs, the global increase in microbial biomass was hypothesized to fuel positive feedbacks of coral mortality through the creation of hypoxic zones by microbial respiration (*Altieri et al., 2017*; *Haas et al., 2016*; *Smith et al., 2006*). The work presented here shows that a physiological transition to overflow metabolisms is the underlying mechanism of bacterial biomass accumulation observed in coral reefs. The loss of oxygen through ebullition facilitates this type of metabolism and adds to the effect of high microbial respiration that depletes oxygen in otherwise oxygen-rich, algae-dominated systems.

### Bacterial overflow metabolism in algae-dominated reefs

The metagenomic results in *Figure 1* show an increase in genes for pathways that shunt carbon to biosynthesis and incomplete oxidation, that is, low $O_2$ consumption (*Russell and Cook, 1995*; *Haas et al., 2016*). The abundance of genes encoding enzymes involved in the Pentose Phosphate (PP) and Entner-Doudoroff (ED) pathways, increased with fleshy algae cover (*Figure 1*). These pathways are alternatives to the canonical glycolytic route for carbohydrate consumption and produce more NADPH than NADH as intermediate electron acceptors. These two reduced coenzymes have distinct cell fates, with NADH preferentially donating electrons to oxidative phosphorylation that consumes oxygen during ATP production (*Pollak et al., 2007*; *Spaans et al., 2015*). Thus, cells utilizing more NADPH pathways consume less $O_2$ relative to carbon compared to cells utilizing more NADH, and store a larger fraction of the consumed carbon into biomass.

The tricarboxylic acid cycle (TCA or Krebs cycle), is a canonically oxidative path for complete decarboxylation of pyruvate. But the TCA cycle can work as a biosynthetic pathway by providing intermediates to amino acids, nitrogenous bases and fatty acids syntheses (*Cronan and Laporte, 2013*; *Sauer and Eikmanns, 2005*). TCA cycle stoichiometry is maintained through the resupply of these intermediates by anaplerotic reactions that use pyruvate (or phosphoenolpyruvate) (*Owen et al., 2002*). In the reefs analyzed here, the abundance of genes encoding anaplerotic enzymes increased, and genes encoding an oxidative decarboxylation step in the Krebs cycle decreased with increasing algal cover. These results indicate a shift towards the use of the Krebs cycle as an anabolic route (*Bott, 2007*; *Sauer and Eikmanns, 2005*). Anaplerotic activity increases in rapidly growing bacteria with high rates of amino-acid synthesis (*Bott, 2007*). These routes shunt organic carbon into biomass accumulation, as opposed to oxidation with electron transfer to $O_2$ (*Russell and Cook, 1995*).

This overflow metabolism is analogous to the Warburg effect, where cells undergoing fast metabolism are limited by enzymes' catalytic rates, as opposed to substrate concentrations (*Vander Heiden et al., 2009*). This type of metabolism is classically described in cancer cells and fast-growing yeast, and consumes large amounts of organic carbon without complete oxidation to $CO_2$, even in the presence of $O_2$. In bacteria, the Warburg effect is a physiological response to changing proteomic demands that optimizes energy biogenesis and biomass synthesis under high energy supply (*Basan et al., 2015*). This biochemical transition occurs because the proteome cost of energy biogenesis by respiration exceeds that of fermentation. A switch to overflow metabolism would benefit heterotrophic bacteria living in a biogeochemical environment of high total DOC but low relative $O_2$ that is observed in algae exudates.

The overflow metabolism predicted by metagenomic analysis was observed in experimental incubations of microbes. When growing on algal exudates, microbes increased cell volume and DOC

consumption per cell (*Figures 2A* and *3*). Yet, these cells displayed the same cell-specific $O_2$ consumption compared to small cells growing on coral exudates (*Figure 2B*). These results indicated that microbes growing on coral exudates fully oxidize the organic carbon consumed, channeling metabolic energy to maintenance costs through NADH- and ATP-producing pathways (*Russell and Cook, 1995*). On algal exudates, bacteria had higher DOC consumption, with a greater fraction of the organic carbon being only partially oxidized and shunted to NADPH and biosynthesis. The deviation of the excess carbon to these pathways caused less $O_2$ consumption (relative to carbon) and increased community biomass (*Figures 2B* and *3*).

We considered and rejected alternative hypotheses for DOC and $O_2$ consumption patterns observed in our experiments, such as *broken* TCA cycles and reactive oxygen species (ROS) detoxification (*Mailloux et al., 2007*; *Steinhauser et al., 2012*). There was no evidence for an increase in genes encoding these pathways in our dataset (*Figure 1—source data 1*). Likewise, the metagenomic transitions observed in situ could not be explained by the rise or disappearance of specific taxonomic groups that changed in abundance with increasing algae cover (*Dinsdale et al., 2008*; *Kelly et al., 2014*). The disconnection between functional and taxonomic profiles is common in microbial communities and has been previously observed in coral reefs (*Kelly et al., 2014*). This discordance is due to strain-level variability in functional genes a result of genomic adaptation to multiple environment selective pressures (*Klingner et al., 2015*; *Martiny et al., 2006*; *Martiny et al., 2009*).

## Overflow metabolism predicts a shift in ecosystem biomass allocation

The cell volume and abundance changes observed in our experiments helps to explain the higher microbial biomass observed in algae-dominated reefs across the Pacific, Caribbean and Indian Oceans (*Haas et al., 2016*). In the Pacific, an increase in total bacterial biomass caused the total bacterial energetic demands to surpass that of macrobes in degraded reefs (*McDole et al., 2012*; *McDole Somera et al., 2016*). The increase in overflow metabolism observed in our dataset provides a mechanism for the increase in biomass and energetic demand observed in degraded reefs, affecting reef-scale trophic interactions (*De'ath and Fabricius, 2010*; *Haas et al., 2016*; *Russell and Cook, 1995*; *Silveira et al., 2015*; *Wilson et al., 2003*).

The changes in biomass observed in situ cannot be attributed to changes in abundance of a single clade. The only clade with an increase relative abundance that could potentially explain the change in cell sizes was *Prochlorococcus*, which has cell volumes of ~0.1 $\mu m^3$ (*Kirchman, 2016*). However, this is the same average cell size observed in the offshore water used as inoculum in our incubation experiments, and the consumption experiments were run in the dark, selecting for heterotrophic microbes (*Figure 3B*). The other groups with significant relationships with fleshy algae cover contributed to only 0.9% to 1.4% of the community (*Figure 1—source data 1*), and their average cell sizes are not consistent with the observed increase in cell size in algae: *Thermotoga* has large cells, but their abundance decreased with algae, *Leuconostoc*, *Rhodopirellula*, and *Methanobacterium* also decreased in abundance with increasing algae. Therefore, none of these clades can explain the change in size observed in this study.

## Ebullition causes $O_2$ loss from coral reefs

Photosynthesis and respiration have a theoretical 1:1 molar ratio of organic carbon and $O_2$ produced and consumed (*Williams et al., 1979*). In holobionts with different proportions of heterotrophic and autotrophic components, such as corals and algae, respiration can consume different fractions of the oxygen and organic carbon produced in photosynthesis (*Tremblay et al., 2016*). The cost of calcification in corals and calcifying algae can also increase the rate of oxygen consumption due to its energetic costs (*Muscatine et al., 1981*). As a result, the DOC:$O_2$ ratios in exudates from calcifying holobionts are predicted to be higher than in fleshy algal exudates. This pattern contradicts experimental data showing higher DOC:$O_2$ ratios in microbe-free algae incubations (*Haas et al., 2011*). Our results suggest that $O_2$ ebullition is the likely cause of this observation. The organic carbon released by algae can be fully solubilized, while a fraction of the $O_2$ nucleates and escapes, leaving behind a high DOC:dissolved $O_2$ ratio. The lack of a relationship between dissolved oxygen concentrations and ebullition rates suggests that the bubbles are formed by heterogenous nucleation on the algae surface (*Freeman et al., 2018*). Differential carbon allocation into biomass can also alter

these ratios, and the quantitative analysis of carbon incorporation using isotope probing in combination with dissolved and gaseous $O_2$ dynamics is the next step to resolve these budgets.

$O_2$ ebullition and bubble injection by hydrodynamics are recognized as a source of error when estimating production in shallow water ecosystems, yet the extent to which ebullition affects these estimates is rarely quantified due to methodological challenges (*Cheng et al., 2014*; *Kraines et al., 1996*). $O_2$ ebullition observed in this study corresponded to 5–37% of net community production. Previous studies quantified that ebullition accounted for the loss of up to 21% and 37% of the $O_2$ production in a salt marsh and a lake, respectively (*Howard et al., 2018*; *Koschorreck et al., 2017*). If in situ bubble nucleation and rise rates are comparable to those in incubations, our results imply that coral reef gross primary production has been significantly underestimated, especially in algae-dominated states (*Howard et al., 2018*). Previous studies reported bubbles on the surface of turf algae, on sediments, and inside coral skeletons colonized by endolithic algae (*Bellamy and Risk, 1982*; *Clavier et al., 2008*; *Freeman et al., 2018*; *Odum and Odum, 1955*). The heterogeneous distribution of bubble nucleation over primary producers entails that benthic community composition determines the magnitude of $O_2$ ebullition on reef-level $O_2$ dynamics. Our model based on the combined results from microbial and primary producer incubations predicts that microbial respiration and ebullition together contribute to the loss of 11.5% to 66.7% of the gross primary production as reef algae cover increases from 0% to 100%. A combination of future in situ incubations and gas exchange studies is required to test these relationships.

## Conclusion

Our study suggests that $O_2$ depletion observed in algae-dominated reefs is a result of high bacterial densities and oxygen ebullition. Ebullition causes a decoupling between photosynthetic fixed carbon and $O_2$, fundamentally changing the biogeochemical environment: $O_2$ loss as bubbles causes an increase in dissolved DOC:$O_2$ ratios, stimulating overflow metabolism in the microbial community (*Figure 6*). These biophysical and physiological changes cause a depletion in oxygen standing stocks that may negatively affect corals and other reef animals.

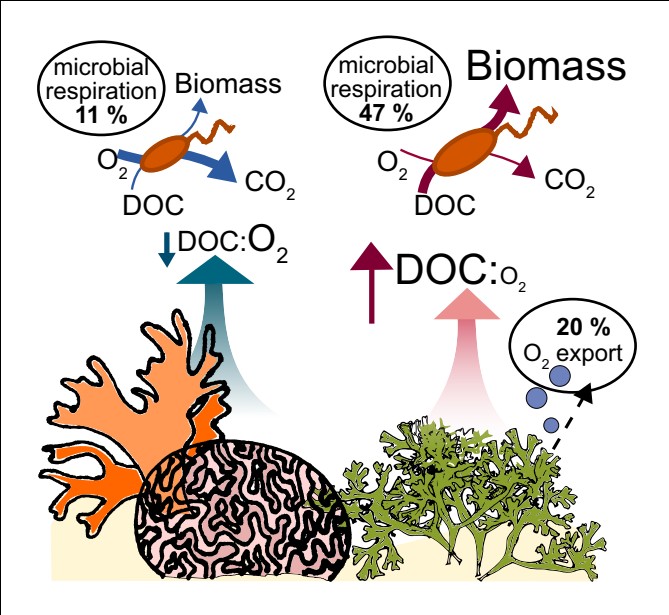

**Figure 6.** Conceptual model of oxygen loss mechanisms from coral reefs. In healthy reefs dominated by calcifying organisms, benthic exudates have low DOC concentrations and low DOC:$O_2$ ratios. The high ratio of electron acceptors ($O_2$) to donors (DOC) stimulates oxidative metabolism in the heterotrophic bacterial community, which sustains low biomass accumulation. In algae-dominated reefs, the loss of oxygen by ebullition leaves behind exudates that are enriched in electron donors (DOC) relative to acceptors ($O_2$). Combined with the higher DOC concentrations, this biogeochemical environment stimulates overflow metabolism in the heterotrophic bacterial community, with incomplete carbon oxidation and accumulation of biomass.

## Materials and methods

### Central carbon metabolism pathways

Microbial metagenomes were sequenced from water samples collected on coral reefs in the Pacific and in the Caribbean. The Pacific samples were collected during NOAA RAMP cruises from 2012 to 2014 to the Hawaiian Islands, Line Islands, American Samoa, and Phoenix Islands. Caribbean samples were collected around the island of Curaçao during the Waitt Institute Blue Halo expedition in 2015. Geographic coordinates for each sampling site are provided in source data 1 for *Figure 1*, along with benthic cover data. At each sampling site, SCUBA divers collected water from within 30 cm of the reef surface using Hatay-Niskin bottles (*Haas et al., 2014a*). One metagenome was generated from each sampling site, and 1 to 5 sites were sampled around each island in the Pacific, and 22 samples from around the island of Curaçao, in the Caribbean. Samples were brought to the ship, filtered through a 0.22 µm cylindrical filter within 4 hr, and stored at $-20°C$ until laboratory processing. DNA was extracted from the filters using Nucleospin Tissue Extraction kits (Macherey Nagel, Germany) and sequenced on an Illumina HiSeq platform (Illumina, USA). Pacific microbial metagenomes described here were previously analyzed for the presence of CRISPR elements, competence genes and Shannon diversity in *Knowles et al. (2016)*. All fastq files from Pacific and Curaçao metagenomes were quality filtered in BBTools with quality score >20, duplicate removal, minimum length of 60 bases, and entropy 0.7 (*Bushnell, 2014*). The 87 metagenomes are available on the NCBI Short Read Archive (PRJNA494971). The number of samples analyzed here surpassed that estimated by power analysis based on the work of *Haas et al. (2016)* (estimated n = 67 for α = 0.05 and β = 0.2 based on the relationship between microbial abundance and algae cover shown in Figure 2B of *Haas et al., 2016*). Relative abundances of genes encoding rate-limiting enzymes in central carbon metabolism pathways were utilized as proxy for the representation of that pathway in the community. The list of pathways and enzymes analyzed here is provided in Supplementary Table 1. Enzyme-specific databases were built using amino acid sequences from the NCBI bacterial RefSeq. BLASTx searches were performed using metagenomic reads against each enzyme gene database, using a minimum alignment length of 40, minimum identity of 60% and e-value $<10^{-5}$. Reads mapping to the database were normalized by total high-quality reads resulting in the percent abundance of each enzyme gene. Relative gene abundances are provided as source data one for *Figure 1*.

The relationship between fleshy algae cover and enzyme gene relative abundances was analyzed by the statistical learning method of supervised random forests regression (*Breiman, 2001*). In the random forest procedure, each tree in the ensemble is grown using a different bootstrap sample of the original data. For the final prediction, the mean prediction regression from all the trees is used. Since each tree is grown with a bootstrap sample of the data, there is approximately one-third of the data that is 'out of sample' for each tree. This out of sample data acts as an internal validation dataset and can be used to determine variable importance for each predictor variable. This is done for each tree by randomly permuting each variable in the out of sample data and recording the prediction. For the ensemble of trees, the permuted predictions are compared with the unpermuted predictions. The magnitude of the mean decrease in accuracy indicates the importance of that variable. Significance of the importance measures is estimated using a permutation test. The random forest easily handles different types of variables and does not require the data to be rescaled or transformed. The random forest consists of an ensemble of decision trees for regression. Predictions, prediction error, and variable importance are estimated from the ensemble of trees. The random forest was grown using 1000 trees and four tree splitting variables. The mean squared error diagnostic plot of the random forest indicated it settled down and therefore enough trees have been used (*Figure 1—figure supplement 1*). The significance of the random forests variable importance was determined by a permutation method, which has an implicit null hypothesis test, and was implemented using the R package rfPermute. The permutation test p-values were obtained using 100 permutation replicates. Variable importance was selected based on the increasing mean squared error, a measure of the importance of a given feature for the correct prediction of a response variable, with alpha cut-off of 0.05. No outliers were removed from any analyses.

The changes in bacterial community composition across the gradient in algae cover were compared to changes in relative abundances of metabolic genes to access the relationship between taxonomy and function. This analysis can indicate whether specific taxa with differences in gene copy number and genome sizes would drive the functional variation. First, the abundance of bacterial

genera in metagenomes was generated through k-mer profiles using FOCUS2 (*Silva et al., 2016*). FOCUS2 computes the optimal set of organism abundances using non-negative least squares (NNLS) to match the metagenome k-mer composition to organisms in a reference database (*Silva et al., 2014*). For annotation at species level, sequences were mapped against a database of bacterial genomes from RefSeq using Smalt at 96% sequence identity (*Ponstingl and Ning, 2010*). The abundance of each organism was computed using the FRAP normalization (Fragment Recruitment, Assembly, Purification), which calculates the fractional abundance of each bacteria in the metagenome accounting for the probability of a read mapping to a genome given the length of both the read and the genomes in the database (*Cobián Güemes et al., 2016*). Taxa with median relative abundance across samples equal to zero were removed from the analysis. The relative abundances of genera and species were separately analyzed through permutation regression random forests with percent algae cover as response variable in both cases. The random forest was grown using 1000 trees and four tree splitting variables. The mean squared error diagnostic plot of the random forest indicated it settled down and therefore enough trees have been used (*Figure 1—figure supplement 1*). Variable importance was selected based on the increasing mean squared error, a measure of the importance of a given feature for the correct prediction of a response variable, with alpha cutoff of 0.05. The complete genomes of the taxa varying in abundance with algae cover deposited in the NCBI RefSeq were manually checked for copy number variation in the genes identified by the functional analysis according the annotations associated with each genome deposited.

## Bacterial DOC and O$_2$ demands

This dark incubation experiment was performed in Mo'orea, French Polynesia. Briefly, exudates from corals and algae released during a light incubation period were 0.2 μm-filtered and inoculated with freshly collected unfiltered forereef seawater to add a compositionally-representative ambient microbial community to the exudate samples. The forereef water column sample contains a microbial community that is most representative of the offshore community that approaches the reef environment in the incoming current (*Haas et al., 2011*; *Nelson et al., 2013*). Inoculation in filtered seawater was utilized as control. All bottles were kept in the dark for 48 hr. DOC, dissolved O$_2$ and microbial abundances were measured at the beginning and at the end of each dark incubation as described above. DOC samples were filtered through pre-combusted GF/F filters (Whatman, 0.7 μm nominal particle retention) and transferred to acid-washed HDPE vials. Samples were kept frozen until analysis according to *Carlson et al. (2010)*. DOC and O$_2$ consumption rates normalized by the bacterial cell yield at the end of each experiment resulted in cell-specific carbon and O$_2$ demands. The fleshy organisms used in this experiment were turf algae and the macroalgae *Turbinaria ornata* and *Amansia rhodantha*, and the calcifying organisms were CCA and the coral *Porites lobata*. Five replicate incubations of each organism were performed (Power analysis estimate n = 4 for α = 0.05 and β = 0.2 based on the incubations of *Haas et al., 2011* and *Haas et al., 2013b*).

## Bacterial cell sizes and biomass

Changes in bacterial abundance, cell size and total biomass in response to primary producers derived from two independent experiments, described below:

Experiment 1: Microbial communities from the reef off the CARMABI Research Station, in the island of Curacao were incubated with four different primary producers: the scleractinian coral *Orbicella faveolata*, CCA, the fleshy macroalgae *Chaetomorpha sp.*, and turf algae. *O. faveolata* colonies were collected at 12 m depth from the Water Factory site (12°10'91' N, 68°95'49' W) and cut into ~10 cm$^2$ fragments. Coral fragments were kept for two weeks at the CARMABI Research Station flow-through tank system. The tank was subject to natural diel light cycles with light intensities comparable to 10 m depth, as measured using HOBO Pendant UA-002–64. CCA, turf, and macroalgae were collected off CARMABI immediately prior to the experiment. Five replicate incubations for each organism were conducted, and five control incubations had no primary producer (Power analysis estimate n = 4 for α = 0.05 and β = 0.2 based on the incubations of *Haas et al., 2011* and *Haas et al., 2013a*). Surface area of organisms is provided in *Figure 3—source data 1*. Incubations lasted for 4 days at 24°C with natural diel light cycles. Bacterial communities at the start and end of the incubation were analyzed by fluorescence microscopy according to *McDole et al. (2012)*. Briefly, cell volume was calculated by considering all cells to be cylinders with hemispherical caps and

individual microbial cell volumes were converted to mass in wet and dry weight using previously established size-dependent relationships for marine microbial communities (*Simon and Azam, 1989*). Differences in cell abundance, cell volume, and total microbial biomass were calculated by the difference between final and initial values, and were normalized by the area of the primary producer in the incubation.

Experiment 2: Five specimens of the coral *Favia sp.* were obtained from a long-term holding tank at the Hawaiian Institute of Marine Biology (HIMB) and placed in independent 5 L polycarbonate containers filled with 0.2 µm-filtered seawater. Five specimens of the fleshy macroalgae *Gracilaria sp.* were collected off HIMB and placed in independent 5 L containers. Additional control containers were filled with filtered seawater. Primary producers were incubated in natural light conditions for 8 hr to release exudates. At the end of the incubation, 2 L of seawater containing exudates were 0.2 µm-filtered and inoculated with 1 L of unfiltered offshore seawater containing water column reef microbial communities. All bottles were incubated for 24 hr in the dark. For microbial abundance and biomass determination, 1 mL samples were collected and analyzed as described above. Differences in cell abundance, cell volume and total heterotrophic microbial biomass were calculated by the difference between final and initial values.

## $O_2$ release by benthic primary producers

The rates of dissolved and gaseous $O_2$ production by benthic primary producers were measured in two tank experiments (POP Experiments 1 and 2). In both experiments, organisms were incubated in custom-made chambers named peripheral oxygen production (POP) bottles. POP bottles are bell-shaped Polyethylene Terephthalate (PET) bottles with a removable base and two sampling ports, one for dissolved analyte sampling, and one at the top for gas sampling. Primary producers were placed at the bottom and bubbles released from their surfaces during incubation accumulated at the top. Primary producers were placed on the base of the POP bottles, inside a larger tank filled with reef water. The bell-shaped container was then placed over the base, and the bottle was sealed. At the end of the incubation, the gas accumulated at the top of the bottles was collected in a syringe and the volume of gas was recorded. The gas was transferred to a wide-mouth container and $O_2$ partial pressure was measured using a polarographic probe (Extech 407510) immediately upon collection. The POP chambers and containers used for measurements were located underwater inside a larger holding tank, so no gas was lost during the quantification procedures. Dissolved $O_2$ was determined using a Hatch-Lange HQ40 DO probe.

POP Experiment 1: The scleractinian coral *Montipora sp.* and the fleshy macroalgae *Chaetomorpha sp.* were collected from coral and macroalgae long-term holding tanks maintained at SDSU. Specimens were collected from the tanks immediately prior to the experiment and placed inside POP bottles. Three specimens of each organism were individually incubated for 2 days with artificial seawater from the coral tank, along with three control bottles under cycles of 12 hr light (150 µmol quanta m$^{-2}$ s$^{-1}$) and 12 hr dark at 27°C.

POP Experiment 2: Four benthic primary producers were analyzed: the scleractinian coral *Orbicella faveolata*, CCA, the fleshy macroalgae *Chaetomorpha* sp., and turf algae. *O. faveolata* colonies were collected at 12 m depth from the Water Factory site in Curaçao (12°10'91' N, 68°95'49' W) and cut into ~10 cm$^2$ fragments. Coral fragments were kept for two weeks at the CARMABI Research Station flow-through tank system. The tank was subject to natural diel light cycles with light intensities comparable to 10 m depth, as measured using HOBO Pendant UA-002–64. CCA, turf, and macroalgae were collected off CARMABI immediately prior to the experiment. Five individual incubations for each organism were conducted. Surface area of organisms is provided in *Figure 3— source data 1*. Incubations lasted for 4 days at 24°C with natural diel light cycles (PAR varied between 150 and 500 µmol quanta m$^{-2}$ s$^{-1}$). The dissolved $O_2$ in macroalgae and CCA bottles were above the accuracy range of the oxygen probe used. In these cases, we used the highest value within the accuracy range of the probe for all following calculations. Therefore, the results shown are lower bounds of the actual $O_2$ concentrations. *Figure 3A* indicates the bottles where supersaturation was observed with gray-filled symbols, and the estimated fraction of oxygen lost by ebullition in *Figure 3C* that were calculated using these lower bound values.

## Open-tank ebullition experiments

Two species of algae (*Chaetomorpha* sp. and *Gracilaria* sp.) and two species of coral (*Montipora* sp. and *Pocillopora* sp.) from the SDSU long-term holding tanks were transferred to incubation tanks immediately prior to each experiment. The surface area of each organism, light intensity (150 to 630 μmol quanta m$^{-2}$ s$^{-1}$) and incubation times are provided in *Figure 5—source data 1*. The experiment started in the morning after a 12 hr period of darkness. Each organism was transferred to a 15 cm x 20 cm x 30 cm glass tank containing seawater from the holding tank. The tank was open on the top to allow free gas exchange with atmospheric air. A GoPro camera was positioned inside the tank facing the organism, and videos were recorded at 4K 30 fps (*Videos 1* and *2* are provided as representative videos of the experiments with *Chaetomorpha* sp. at 150 and 300 PAR, respectively). Additional pictures were taken with a Canon RebelT3 with 18–55 mm lens to obtain bubble sizes. Dissolved O$_2$ was measured throughout the experiments using a Hatch-Lange HQ40 DO probe. The number of bubbles produced per minute was determined by manual counts of bubbles rising from the algae in the video. Bubble sizes were determined from a total of 160 bubbles (at least 40 bubbles from each experiment) from the still images in Adobe Illustrator.

## Estimates of oxygen loss by ebullition

The internal pressure, $p_i$, of a bubble was estimated using the Laplace's law using a standard surface tension model, $p_i = p_e + 2\sigma_w/R_b$ (*Bowers et al., 1995*). Here, $p_e$ was the external pressure (assumed at atmospheric pressure, $p_e = 1\ atm$), $\sigma_w$ the surface tension of water (71.99 ± 0.05 mN/m at 25°C, *Pallas and Harrison, 1990*), and $R_b$ the radius of the bubble. The moles of oxygen in a bubble, $n_b$, were estimated using the ideal gas law, $n_b = (p_i V_b)/(RT)$, which is a good approximation for oxygen at normal conditions (*Christensen et al., 1992*). Here, $R$ was the universal gas constant (8.314 J K$^{-1}$ mol$^{-1}$), $T$ the temperature (25°C or 298.15 K), and $V_b$ was the volume of the bubble, which was approximated as spherical, $V_b = (4\pi/3)R_b^3$. The rate of bubble production necessary to generate the moles of oxygen in the gas phase of the POP experiment, $n_{POP}$, was obtained by assuming a constant bubble production: $r_b = N_b/t_{light}$. Here $N_b = n_{POP}/n_b$ was the number of bubbles needed, and $t_{light}$ was the total time of light exposure in the POP experiments (2 to 4 days of 12 hours of light per day). The rate was normalized by the covered area ($A_p$) of the primary producer, $\rho_b = r_b/A_p$.

The estimated ebullition rate was obtained for macroalgae in the POP experiment using the surface values ($A_p$) and moles ($n_b$) values reported in *Figure 4—source data 1*. The bubble diameter utilized was derived from the open tank ebullition experiment. The mean, first quartile (25%) and third quartile (75%) of bubble sizes were used to obtain the range of ebullition rates necessary to recover the gas in headspace of the POP experiments, indicated by the shaded area in *Figure 5A*.

## Statistical tests

All statistical analyses were performed using the software R. The response variables from the incubation experiments were tested for normality using the Shapiro–Wilk test. Due to lack of normality (Shapiro–Wilk, p<0.05), the non-parametric Kruskal-Wallis test was used to test if there were differences in treatments followed by the post-hoc Wilcoxon test with the False Discovery Rate (FDR) multiple-comparison correction with a significance cutoff of p<0.05. Because gaseous O$_2$ production and fraction of total O$_2$ in the form of gas were not significantly different between POP Experiments 1 and 2 (Kruskal-Wallis, p>0.05), the results of both experiments were combined in subsequent tests. Statistical tests comparing calcification functional groups were performed by combining corals with CCA as calcifying and turf algae with fleshy macroalgae as fleshy organisms, based solely on the presence of absence of calcification. No outliers were removed.

## Uncertainty quantification

All measurements from incubations had an associated error originating from probe accuracy, organism surface area measurement software, volume in incubation chambers, incubation time and microscope resolution. The uncertainty quantification of rates calculated using combination of measurements was performed based on error propagation (*Taylor, 1997*). For error generated from two or more variables, the derivative of the source errors were applied (*Rice, 2006*). All uncertainty propagated from the device measurements were at least five times smaller than the statistical standard deviations from the experiments, except for control incubations where the statistical variation

is in the same order as the uncertainty, as expected (*Figures 2*, *3* and *4* source datasets). Therefore, uncertainty was not incorporated in statistical comparisons between treatments.

## Predictive model

The relative contribution of microbial respiration and ebullition to the removal of oxygen in an idealized reef was performed using a weighted linear model. The per-cell respiration rates from the dark experiment (*Figure 2—source data 1*) were applied to the bacterial abundances sustained by one decimeter square ($dm^2$) of primary producer observed in POP experiment 2 (*Figure 3—source data 1*). These rates were scaled up to a $1\ m^3$ reef by normalizing the rate by the percent cover of coral and fleshy algae on the benthos (*Figure 5—source data 2*). The model focused on these two primary producer groups due to the availability of data on their ebullition rates. The derivation of the model is provided below.

The production of $O_2$ in the gas phase per $m^2$ was calculated from the POP experiments for corals, $\rho_{coral}^{gas}$, and fleshy algae, $\rho_{coral}^{gas}$. The net production of dissolved $O_2$ per area ($m^2$) was also obtained from the POP experiments for coral, $\rho_{coral}^{net}$, and fleshy algae, $\rho_{alga}^{net}$. The microbial respiration rate of dissolved $O_2$ per cell was obtained from the dark incubation experiment (*Figure 2*). The microbial concentration from the POP experiment at the end point was assumed as the reference value to obtain the rate of microbial respiration per volume ($m^3$) in the POP experiment. This was obtained for coral, $\Omega_{coral}^{microb}$, and fleshy algae, $\Omega_{alga}^{microb}$, incubations. To estimate the gross production rate of dissolved oxygen in a reef, a boundary layer of a meter was considered, and above this layer the influence of the primary producers on the oxygen content was neglected (*Barott and Rohwer, 2012*; *Shashar et al., 1996*). The gross production of dissolved $O_2$ per meter square of primary producers was, thus, estimated from the coral incubations, $O_{coral}^{diss} = \rho_{coral}^{net} \times 1\ m^2 + \Omega_{coral}^{microb} \times 1\ m^3$, and the fleshy algae incubations, $O_{alga}^{diss} = \rho_{alga}^{net} \times 1\ m^2 + \Omega_{alga}^{microb} \times 1\ m^3$. The gross production rate of $O_2$ was obtained by combining the dissolved oxygen and gas oxygen obtained from the coral, $O_{coral}^{gross} = O_{coral}^{diss} + \rho_{coral}^{gas} \times 1\ m^2$, and algae, $O_{alga}^{gross} = O_{alga}^{diss} + \rho_{alga}^{gas} \times 1\ m^2$, experiments. The benthos of an ideal reef was assumed to be covered by a combination of coral and algae, $C + A = 1$, where $C$ was the fraction of coral coverage and $A$ the fraction of algal coverage. The gross production rate of $O_2$ in a reef was estimated by the linear weighted model $O_{reef}^{gross} = C \cdot O_{coral}^{gross} + A \cdot O_{algae}^{gross}$. The production rate of $O_2$ gas, $O_{reef}^{gas}$, and microbial consumption rate of $O_2$, $O_{reef}^{microb}$, in the reef were calculated analogously. The coral coverage fraction was expressed in terms of algal coverage fraction, $C(A) = 1 - A$, to investigate the impact of algal coverage in the oxygen budget of the reef. The fraction of oxygen loss in ebullition was defined as $E = O_{reef}^{gas}/O_{reef}^{gross}$. In terms of the algal coverage, the ebullition fraction was given by the shifted Hill function of order one

$$E(A) = \frac{\left(\frac{\Delta O^{gas}}{\Delta O^{gross}}\right)A + \frac{O_{coral}^{gas}}{\Delta O^{gross}}}{\frac{O_{coral}^{gross}}{\Delta O^{gross}} + A}$$

Here, $\Delta O^{gross} = O_{alga}^{gross} - O_{coral}^{gross}$ and $\Delta O^{gas} = O_{alga}^{gas} - O_{coral}^{gas}$. A similar expression was obtained for the fraction of oxygen rate consumed by microbes as a function of the fraction of algal coverage, $M(A)$, where the gas terms above become microbial terms.

## Acknowledgements

We thank the crew and captains of the NOAA ship Hi'ialakai and the Waitt Institute ship Plan B that contributed to sampling. We thank the CARMABI and HIMB for the support conducting field experiments. We thank Robert Edwards at SDSU for access to computer servers funded by the NSF (CNS-1305112 to Robert A Edwards). This work was funded by the Gordon and Betty Moore Foundation (grant 3781 to FR) and Spruance Foundation. CBS was funded by CNPq (234702) and Spruance Foundation. TNFR was supported by the NSF (G00009988).

# Additional information

## Funding

| Funder | Grant reference number | Author |
|--------|------------------------|--------|
| National Science Foundation | G00009988 | Ty N F Roach |
| Gordon and Betty Moore Foundation | 3781 | Forest Rohwer |
| Conselho Nacional de Desenvolvimento Científico e Tecnológico | 234702 | Cynthia B Silveira |
| Spruance Foundation | | Cynthia B Silveira |

The funders had no role in study design, data collection and interpretation, or the decision to submit the work for publication.

## Author contributions

Cynthia B Silveira, Conceptualization, Data curation, Formal analysis, Supervision, Validation, Investigation, Visualization, Methodology, Writing—original draft, Writing—review and editing; Antoni Luque, Formal analysis, Investigation, Writing—review and editing; Ty NF Roach, Andreas Haas, Conceptualization, Investigation, Writing—review and editing; Helena Villela, Adam Barno, Kevin Green, Esther Rubio-Portillo, Investigation, Writing—review and editing; Brandon Reyes, Mark Hatay, Investigation, Methodology; Tram Le, Spencer Mead, Investigation; Mark JA Vermeij, Resources, Investigation, Writing—review and editing; Yuichiro Takeshita, Conceptualization, Writing—review and editing; Barbara Bailey, Data curation, Formal analysis, Writing—review and editing; Forest Rohwer, Conceptualization, Resources, Supervision, Investigation, Methodology, Writing—review and editing

## Author ORCIDs

Cynthia B Silveira (iD) https://orcid.org/0000-0002-9533-5588

## Decision letter and Author response

Decision letter https://doi.org/10.7554/eLife.49114.sa1
Author response https://doi.org/10.7554/eLife.49114.sa2

# Additional files

## Supplementary files

• Supplementary file 1. List of genes encoding rate-limiting enzymes participating in central carbon metabolism, reactive oxygen species detoxification and replication/translation analyzed in this study. Gene abundances are provided in *Figure 1—source data 1*.

• Transparent reporting form

## Data availability

All experimental data generated and analyzed in this study are included in the supporting files and the metagenomic data is available on the NCBI SRA under PRJNA494971.

The following dataset was generated:

| Author(s) | Year | Dataset title | Dataset URL | Database and Identifier |
|-----------|------|---------------|-------------|-------------------------|
| Cynthia Silveira | 2018 | Reef Metagenomes | https://www.ncbi.nlm.nih.gov/search/all/?term=PRJNA494971 | NCBI Sequence Read Archive, PRJNA494971 |

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
