## [Decision Letter]

**Acceptance summary:**

Overfishing, eutrophication, and climate change are shifting the composition of many coral reefs towards more algae and less coral. Algal have greater net oxygen production than corals, yet paradoxically this shift in producers decreases ecosystem oxygen availability while stimulating bacterial growth, a process known as microbialization. Silveira and coauthors combine metagenomic field data and laboratory incubations of algae, corals, and reef bacteria to tell a novel and compelling story: 20% of the oxygen generated by reef algae could escape to the atmosphere in bubbles, leaving behind excess dissolved organic carbon. This imbalance stimulates metabolic pathways in heterotrophic bacteria that tilt the cellular scales towards greater organic carbon uptake and biomass accumulation, and thus greater ecosystem respiration. Taken together, these processes suggest a mechanistic underpinning for the observed ecosystem changes accompanying microbialization. The authors' findings reinforce the growing body of oceanographic and limnologic work indicating that photosynthetically stimulated bubbling can lead to underestimation of oxygen-based aquatic ecosystem metabolism fluxes. This work should further prompt in situ investigations of how bubbles and alternative metabolic pathways can lead to non-canonical relationships between carbon and oxygen fluxes and stimulate ecosystem shifts-these may be important factors driving the response of shallow, productive aquatic environments to changing climate.

**Decision letter after peer review:**

Thank you for submitting your work entitled "Biophysical and physiological processes causing oxygen loss from coral reefs" for consideration by *eLife*. Your article has been reviewed by two peer reviewers, and the evaluation has been overseen by a Senior Editor in consultation with an outside expert. The reviewers have opted to remain anonymous.

Our decision has been reached after consultation between the reviewers. Based on these discussions and the individual reviews below, we regret to inform you that your work will not be considered further for publication in *eLife*.

The work is clearly very interesting and we all were convinced that the main thesis could well be right, but that at this stage, the data does not fully support your conclusions. In addition I wanted to note three points from my own reading of the work and the reviews, that might help you prepare for the next steps for this work:

1) The arguments that the microbial community uses less energy-efficient carbon consumption pathways is based on metagenomic analyses. However, metagenomic analyses alone are not sufficient for identifying what a microbial community is really doing, they only reveal the metabolic capabilities of the community. Transcriptomic or proteomic analyses would be essential to show that there was a shift of the community in their use of carbon consumption pathways.

2) It was not clear which advance the metagenomic analyses of this paper provides as compared to the advances presented in the earlier paper by Haas et al., 2016,using a smaller dataset of 60 coral reef sites (vs. 87 here). Was the same data set used as in Haas et al. albeit expanded by 27 additional sites?

3) As noted by the first reviewer, the statistical analyses require much additional work.

Reviewer #1:

I think the datasets presented in this manuscript are promising and could ultimately support the stated conclusions. However, this manuscript has multiple issues that I do not believe can be adequately addressed within the scope of this submission (i.e. without a further round of peer review).

In summary, the major issues I identify include:

1) Shortcomings with the metagenomics approach and related statistics: The authors have set themselves a harder task by using genomic rather than transcriptomic data, and haven't convincingly shown that the genomic data can answer their question about activity of distinct metabolic pathways. The choice of a classification technique in place of standard statistical tools that work well with such data is not explained by the authors, and the details provided are insufficient to show that the result is statistically robust.

2) Methodological and analysis errors with the oxygen ebullition experiments: The oxygen data does not appear to have undergone QA/QC, leading to spurious results based on readings entirely beyond the accurate range of the sensor, and in many cases beyond the sensor detection limits. There is no uncertainty analysis of resulting rates. The values presented are extremely supersaturated with respect to dissolved oxygen and characteristic of equilibrium conditions in the gas phase, meaning the results are unlikely to be environmentally relevant and may not be related to ebullition.

3) Ratios of DOC and oxygen uptake in incubations which are not consistent with the expectations laid out in the Discussion: The DOC and oxygen uptake rates are similar to the expected 1:1 ratio for fleshy algae exudates, while it is the control and coral experiments that have unexpected ratios. This implies that the expectation value presented is incorrect (because the rate of DIC increase is not equal to the rate of DOC uptake), that an unexplained process is occurring in the coral exudate and control samples, or that the uncertainties are substantial (there was again no uncertainty analysis of the rates in this experiment).

4) Lack of discussion as to what composes the bacterial community in incubation experiments and how that informs or caveats interpretation of the experimental results: Without discussion of what is present in the experiments it is hard to interpret those results. Is community composition changing? Are certain clades dominating the observed responses? Does the implicit choice of community structure in the incubations bias the results towards certain outcomes (e.g. a community that normally grows near algae has a stronger response to algal exudates than to a carbon source it is not adapted for?).

A longer explanation of these concerns (with specific details and some suggestions for ways forward) follows, with the idea that it may be useful to the authors should they choose to resubmit.

1) Metagenomics a) Metagenomics seems like an indirect tool to answer the question the authors have posed regarding gene expression as a function of algal cover. In the absence of transcriptomic data that identifies active gene expression, some logical leaps are required to connect gene abundance to active metabolic pathways in the microbial community, and the current paper does not provide sufficient detail to outline these or convince a reader that they are justified.

b) The statistical analysis of the metagenomic data is not adequately described for me to evaluate how robust it is. In particular there is no indication that the genetic data was normalized to bacterial abundance or some other sensible scheme to avoid biases due to very different sampled communities (dividing by the number of good reads is insufficient). It is unclear what if any null hypothesis was tested. 14% covariance between differences in the gene community and algal cover is a weak signal given that the five genes were explicitly chosen with the expectation of showing a signal, and at the chosen alpha value (0.05) 5% of results are false positives by definition. Using a random forest approach, the authors might evaluate the null hypothesis based on the distribution of variance explained over ~1000 random subsets of five genes, and then evaluate the significance of the 14% value relative to that null distribution (which in my experience is unlikely to have both mean and standard deviation equal to 0% as appears to be implicit in the current analysis).

c) I question why a random forest classification and regression approach was used when something like an ANOVA should work well and is commonly used with -omics data (with clearly testable hypotheses, or a PCA if not). Since a random forest approach was used, more details are required about the number of trees and tree size, the division of the data in training, validation, and test subsets and any resulting tuned hyperparameters, and finally whether stated errors refer to out of bag (classification) or misfit (regression) errors. My personal experience is that the default statistical software settings for random forests can lead to non-robust results, thus care must be taken in the selection (and description) of these characteristics.

d) Since the metagenomes are already assembled the authors should know which species/strain level bacteria are present. Thus I would think a more convincing assessment of the effect of copy number variability would be to look at orthologs and paralogs in at the species/strain level and then evaluate if certain extremes of algal cover (very low, very high) are associated with particular species with many gene replicates or deletions that drive the overall trend. Doing this at the genera level as is the current choice could mute any problematic signal, unless the genera is monospecific. The manuscript states, however, that there is in fact high strain level variation, meaning that this choice is potentially masking a problem. At a minimum, a good reference should justify the choice. And this analysis still does not address the potential complication of variability in gene expression between sites. Finally, from Supplementary file 1 it appears that autotrophic *Prochlorococcus* has a stronger relationship with algal cover than the heterotrophic bacteria that are the focus of this work. It may be necessary to confirm *Prochlorococcus* (and *Synechococcus*, etc.) are not obfuscating your signal.

2) Oxygen ebullition experiments a) Based on the Figure 4—source data 1, it appears that there was a failure to do QA/QC on the oxygen concentrations in the POP experiments. The dissolved oxygen concentration at the end of all of the non-control experiments are outside the stated accurate range for the instrument used, 0-20 mg/L or 0-200% saturation, whichever is lower (https://www.hach.com/hq40d-portable-multi-meter-ph-conductivity-salinity-tds-dissolved-oxygen-do-orp-and-ise-for-water/product-details?id=7640501639). Since this is an optode based optical system, the concentration based range is more notion than reality and the saturation state should always be the limiting bound at high oxygen conditions. Optode based dissolved oxygen sensors have nonlinear responses to concentration, so the deviation from the true value may vary in a way that is not readily predictable (even within the 0-200% range the accuracy varies substantially). Not only are all the experiments beyond the accurate range, many of them (including all the CCA experiments) are beyond the instrument bounds (the internal QA filter is apparently set to 30 mg/L given the constant endpoint values of ~935 μm in many experiments).

b) Hypothetically the above experimental data from below the upper instrument detection bound could be salvaged, if a full range calibration to Winkler titrations or another sensor were to be run on the same sensor cap originally used, at each of the experimental temperatures (the oxygen sensitivity of the optode foil is temperature dependent). See Bushinsky and Emerson, 2013 for an example of what this might look like (https://doi.org/10.1016/j.marchem.2013.05.001). But the sensor characteristics have a substantial "grow-in" if they are new, and change if they are dry for an extended period, so this would have the best results if the sensor cap had already seen several months of use prior to the experiments and has been kept continuously moist since. But without the beyond detection samples there is insufficient experimental power to determine significance…

c) I'm concerned with how the sampling of the headspace gas is described. The manuscript says the gas was transferred from a syringe to a wide mouthed container that the second probe was placed into. Gas diffuses quickly, so I don't see how this could lead to accurate measurements-but I may be misunderstanding, and this refers to the dissolved sample? At any rate this description needs to be clarified.

d) Setting aside the accuracy issues above, all of the headspace values are approximately in equilibrium with the dissolved oxygen you measured. Henry's law solubility coefficients predict concentrations of 0.36-0.94 mg/mL and 1.3-20.6 mg of oxygen in the gas phase above the water samples-close to the 0.4-0.85 mg/mL and 1.1-17 mg O_2_ measured (assuming 1 atm pressure, and neglecting any lag between production and diffusion). Thus I think that the presence of a headspace isn't proof of the importance of ebullition in a natural environment-it is proof that a highly supersaturated solution will push gas into a headspace (like CO_2_ is in a non-experimental pop bottle). I do suspect bubble nucleation was essential, since the controls were also supersaturated and formed no headspace, but they were a lot less supersaturated so this would be more convincing with evidence of rising bubbles in the chambers-otherwise thermodynamics alone is sufficient without any biophysical interactions. At any rate what the above suggests to me is these chambers are unlikely to reflect environmental outcomes regarding the amount of oxygen released in bubbles. In this light it is very hard to support the statement that in algal dominated reefs 1/3 of the oxygen escapes as bubbles.

e) There is no uncertainty analysis of these rates. Given the relatively coarse measurements of bubble volume (to nearest 0.5 mL), short durations (2-4 hours), and uncertainties in saturation state (even when the instrument is within its specified range), both the gas production and dissolved oxygen change must have significant uncertainties, but these are not quantified. I think they should be, and relative uncertainty of each experiment shown so a reader can gauge how significant differences between the box and whisker categories actually are. Also, 2 pts don't make for a convincing rate.

3) DOC and oxygen uptake experiments

The DOC and oxygen uptake ratios in these experiments are far from 1:1, but not in the manner described to support arguments about less complete oxidation of organic matter. To use respiratory quotients with DOC it must be assumed that DOC uptake is equivalent to CO_2_ (DIC) release. The arguments put forward in this manuscript about the incomplete oxidation of organic matter would argue against this for the macroalgae, but I'll set that aside for the moment. At any rate, the -DOC/-O_2_ ratio for the fleshy algae exudate is ~0.9, right in line with theoretical and measured values of +DIC/-O_2_ for bacterioplankton (Robinson 2019, https://doi.org/10.3389/fmars.2018.00533). It is instead the controls and corals that seem to have unexpected values on the order of 0.3. So I see three options: 1) the -DOC/-O_2_ rate ratio shouldn't actually be ~1:1 (or rather 0.9), and thus the discussion of respiratory quotients is a red herring. 2) the corals and controls are doing something unusual that requires an explanation. 3) There are major biases in the rates. I think the error analysis I recommended for the O_2_ production fluxes should be done here as well and could inform this question. So overall I agree that the incubations show more oxygen consumption per unit of DOC consumed with the control and coral exudates than with the algal exudates, but since none of these categories meet the stated expectations this needs to be explored in more detail to confirm that the result is real and robust.

4) Bacterial community structure

The bacterial community structure is not discussed. How it diverges across sampled environments is challenging to ascertain in Figure 1. I don't find any information on what the bacterial community looked like within the various incubation experiments. Thus I can't evaluate whether the incubation results are a function of, e.g., incubating a community adapted to one particular environment in a way that would bias the experimental outcomes. While there is a mention about changing community structure with regards to the incubations (Discussion), there is not data in the supplement regarding the community in these experiments, nor would I expect changing genetic composition over the course of experiments lasting hours to days (again, a question for transcriptomics?). This makes it hard to interpret what is happening in the experiments where change in biomass is evaluated. Are changes in biomass due primarily to a single clade? Are certain larger types of bacteria being selected for, or are particular bacteria growing larger while others do not? Is this an expected result based on where those bacteria normally are environmentally successful?

Reviewer #2:

This study provides a theoretical and experimental based evidence approach to explain the observations for O_2_ depletion in algal dominated coral reefs. The processes of ebullition and microbial metabolism are postulated to drive this lower observed oxygen level in algal dominated reefs. Overall the manuscript is very interesting and details a nice combination of environmental genomics based analysis linked with controlled experimental trials. The postulated physical and biological reasons for reduced O_2_ levels are supported by the experimental data, though the conclusions are still a little speculative. However this approach represents a great approach to hypothesis driven science; and while not able to definitely confirm these process (i.e. ebullition and microbial metabolism) as driving O_2_ depletion, it sets up the concepts to be further tested in environmental settings. Hence the manuscript is worthy of publication with a few comments detailed below highlighting potential areas for improvement.

[Editors’ note: what now follows is the decision letter after the authors submitted for further consideration.]

Thank you for choosing to send your work entitled "Biophysical and physiological processes causing oxygen loss from coral reefs" for consideration at *eLife*.

We thank you for sharing your rebuttal letter. We would welcome a revision of your work for a new cycle of review and note that the revision should address the following concerns:

The text is missing or unclear on the rationale for a number of the statistical and analytical choices, and has weaknesses in some of the lines of evidence, including unresolved problems with the oxygen production experiments. Thus the current manuscript, in the opinion of two reviewers, presents the interpretations with a certainty and forcefulness which we did not think is warranted by the present constellation of suggestive (but not definitive) evidence.

Thank you for clarifying the oxygen units. However, all of the CCA and most of the fleshy macroalgae experiments were beyond the detection bounds of your instrument (20 mg/L), as evidenced by the constant endpoint concentrations (937.5 umol/1.5L=625µM=20 mg/L) across several nominally independent experiments. The headspace concentrations also still appear to reflect equilibrium conditions with the underlying water when the solution is strongly supersaturated (as it is in most of the non-control experiments). How relevant are these experiments for explaining oxygen loss under realistic environmental conditions (less supersaturated and open to the atmosphere)?

There are statistical techniques for dealing with non-normally distributed data (or more accurately, non-conditionally normal error distributions-A random forest regression may well be a good choice for this analysis, but a rationale should be explained, and the particulars of how it is used clearly stated.

Whether 5 or 16 genes, the correlations to environmental variables need more context. Our concern remains about the effect size versus a null hypothesis of some other random set of genes from your already sequenced genomes, which are far larger than 16 genes. The details required to assess what differences are "real" are currently insufficient. e.g. was the data normalized to bacterial abundance? A single 100mb genome will have far more reads than a 1mb genome, even if there are 10 times more of the smaller-genome organism. You may well have made considered choices for these and other potential issues in your analysis, but you tell the reader none of this, even in the supplement.

Regarding your choice of approach for the metagenomic data. Please provide context and defend the choice (and aspects of the analysis of this data) in the text. It would be a good choice to highlight the rationale of this approach in as succinct a manner in the text as you do in your rebuttal letter.

We look forward to your revision.

---

## [Author Response]

[Editors’ note: the author responses to the first round of peer review follow.]

The work is clearly very interesting and we all were convinced that the main thesis could well be right, but that at this stage, the data does not fully support your conclusions. In addition I wanted to note three points from my own reading of the work and the reviews, that might help you prepare for the next steps for this work:1) The arguments that the microbial community uses less energy-efficient carbon consumption pathways is based on metagenomic analyses. However, metagenomic analyses alone are not sufficient for identifying what a microbial community is really doing, they only reveal the metabolic capabilities of the community. Transcriptomic or proteomic analyses would be essential to show that there was a shift of the community in their use of carbon consumption pathways.2) It was not clear which advance the metagenomic analyses of this paper provides as compared to the advances presented in the earlier paper by Haas et al., 2016, using a smaller dataset of 60 coral reef sites (vs. 87 here). Was the same data set used as in Haas et al. albeit expanded by 27 additional sites?3) As noted by the first reviewer, the statistical analyses require much additional work.Reviewer #1:I think the datasets presented in this manuscript are promising and could ultimately support the stated conclusions. However, this manuscript has multiple issues that I do not believe can be adequately addressed within the scope of this submission (i.e. without a further round of peer review).In summary, the major issues I identify include:1) Shortcomings with the metagenomics approach and related statistics: The authors have set themselves a harder task by using genomic rather than transcriptomic data, and haven't convincingly shown that the genomic data can answer their question about activity of distinct metabolic pathways.

Previous studies have shown the fast genomic adaptation of offshore bacterial communities to the reef environment – hours to days, the timescale of residence time – and the validity of using metagenomics to describe functional adaptation of bacteria (Nelson et al., 2013, Wegley-Kelly et al., 2015). This genomic adaptation occurs at the strain level, with rapid selection of strain variants with highest fitness under different selective pressures (Martiny, 2006, 2008). Therefore, the metagenomic analysis is a strong tool to indicate which carbon metabolic pathways are selected in the reef environments. This analysis provided a hypothesis that was experimentally validated in our in vitro experiments, i.e., direct quantification of bacterial activity. Transcriptomics is much more variable with and, just like metagenomics, only demonstrates potential and not activity. This argument is described in the first paragraph of the subsection “Abundance of carbon metabolism genes across algal cover gradient”.

The choice of a classification technique in place of standard statistical tools that work well with such data is not explained by the authors, and the details provided are insufficient to show that the result is statistically robust.

In our study, we used a regression random forest analysis (not classification – subsection “Abundance of carbon metabolism genes across algal cover gradient”, second paragraph) with algae cover as response variable, and a permutation test with corresponding p-values. Deciphering the relationships between microbial community composition, function, and environmental variables is a statistically challenging task due to the multi-dimensionality of these datasets. This challenge can be addressed by applying machine learning tools, such as random forests (Thompson et al., 2019, Chang et al., 2017, Edoardo et al., 2016, Hunter et al., 2016). The use of more standard multiple linear regression, inference, and model selection requires normality assumptions, which are not needed for a random forest. Since a random forest is a tree-based method it has the advantage that it can be implemented with a large number of predictor variables, and the variable importance ranks obtained from random forest may be used as a variable selection tool. We have included a paragraph in the Results and expanded the Materials and methods describing the random forest methodology:

“The relationship between fleshy algae cover and enzyme gene relative abundances was analyzed by the statistical learning method of supervised regression random forests (Breiman, 2001). […] Variable importance was selected based on the increasing mean squared error, a measure of the importance of a given feature for the correct prediction of a response variable, with α cutoff of 0.05. No outliers were removed from any analyses.”

2) Methodological and analysis errors with the oxygen ebullition experiments: The oxygen data does not appear to have undergone QA/QC, leading to spurious results based on readings entirely beyond the accurate range of the sensor, and in many cases beyond the sensor detection limits. There is no uncertainty analysis of resulting rates. The values presented are extremely supersaturated with respect to dissolved oxygen and characteristic of equilibrium conditions in the gas phase, meaning the results are unlikely to be environmentally relevant and may not be related to ebullition.

The reviewer mistook the values presented in Figure 4—source data 1: these values are not concentrations (µM), but actual number of mols, accounting for the volume of the incubation chamber, 1.5 L. We used calibrated probed, and performed an uncertainty quantification now included in the Source datasets provided with the manuscript. In the cases where dissolved O_2_ was above the 20 mg/L upper limit, we conservatively used the value 20 mg/L. To test whether ebullition is environmentally relevant we performed additional open-tank experiments (Figure 5—source data 1).

3) Ratios of DOC and oxygen uptake in incubations which are not consistent with the expectations laid out in the Discussion: The DOC and oxygen uptake rates are similar to the expected 1:1 ratio for fleshy algae exudates, while it is the control and coral experiments that have unexpected ratios. This implies that the expectation value presented is incorrect (because the rate of DIC increase is not equal to the rate of DOC uptake), that an unexplained process is occurring in the coral exudate and control samples, or that the uncertainties are substantial (there was again no uncertainty analysis of the rates in this experiment).

The reviewer states that “to use respiratory quotients with DOC it must be assumed that DOC uptake is equivalent to CO_2_ (DIC) release”. However, a fraction of the DOC consumed by a microbial cell is not oxidized to DIC, but instead allocated into biomass. This fraction greatly varies with cell physiology as reviewed in Russel and Cook, 1995. Therefore, we disagree with the reviewer’s statement that DOC:O_2_ ratios are equivalent to respiratory quotients.

4) Lack of discussion as to what composes the bacterial community in incubation experiments and how that informs or caveats interpretation of the experimental results: Without discussion of what is present in the experiments it is hard to interpret those results. Is community composition changing? Are certain clades dominating the observed responses? Does the implicit choice of community structure in the incubations bias the results towards certain outcomes (e.g. a community that normally grows near algae has a stronger response to algal exudates than to a carbon source it is not adapted for?).

The bacterial community utilized in all experimental conditions was the same: forereef communities, which represent offshore microbes that have not been exposed to the reef flat biogeochemistry and, therefore, were not yet selected or subject to implicit choice of community structure (subsection “Bacterial DOC and oxygen demands”). The bacterial community composition could not explain the trends in gene abundance (Results: subsection “Abundance of carbon metabolism genes across algal cover gradient”; Discussion: subsections “Bacterial overflow metabolism in algae-dominated reefs” and “Overflow metabolism predicts a shift in ecosystem biomass allocation”). None of the taxa that increased with algae cover consistently encode multiple copies of the genes studied, or lack these genes.

A longer explanation of these concerns (with specific details and some suggestions for ways forward) follows, with the idea that it may be useful to the authors should they choose to resubmit.1) Metagenomics a) Metagenomics seems like an indirect tool to answer the question the authors have posed regarding gene expression as a function of algal cover. In the absence of transcriptomic data that identifies active gene expression, some logical leaps are required to connect gene abundance to active metabolic pathways in the microbial community, and the current paper does not provide sufficient detail to outline these or convince a reader that they are justified.

As discussed above, previous studies have shown the fast genomic adaptation of bacterial communities to environmental change – residence time – and the validity of using metagenomics to describe functional adaptation of bacteria (Nelson et al., 2013, Kelly et al., 2014). Additionally, the metagenomic analysis was used here as a hypothesis-generating tool, which was then tested using controlled experiments of bacterial physiology.

b) The statistical analysis of the metagenomic data is not adequately described for me to evaluate how robust it is. In particular there is no indication that the genetic data was normalized to bacterial abundance or some other sensible scheme to avoid biases due to very different sampled communities (dividing by the number of good reads is insufficient). It is unclear what if any null hypothesis was tested. 14% covariance between differences in the gene community and algal cover is a weak signal given that the five genes were explicitly chosen with the expectation of showing a signal, and at the chosen alpha value (0.05) 5% of results are false positives by definition. Using a random forest approach, the authors might evaluate the null hypothesis based on the distribution of variance explained over ~1000 random subsets of five genes, and then evaluate the significance of the 14% value relative to that null distribution (which in my experience is unlikely to have both mean and standard deviation equal to 0% as appears to be implicit in the current analysis).

The data actually consists of 87 independent metagenomes (subsection “Abundance of carbon metabolism genes across algal cover gradient”, first paragraph) and 23 genes of rate-limiting steps in central carbon metabolism and control genes (see the second paragraph of the aforementioned subsection and Supplementary file 1). Fleshy algae cover was the response variable of the random forest regression analysis. We have included more details on the number of trees grown in the forest, the examination of the MSE, and the permutation test, which incorporates a null hypothesis test (see the aforementioned subsection and subsection “Central carbon metabolism pathways”, second paragraph, diagnostic plots are shown in Figure 1—figure supplements 2 and 3). The abundance of bacterial taxa and the potential contributions of taxa with different genomes sizes was analyzing using FRAP (Cobian-Guemes et al., 2016) (–subsection “Central carbon metabolism pathways”, last paragraph).

c) I question why a random forest classification and regression approach was used when something like an ANOVA should work well and is commonly used with -omics data (with clearly testable hypotheses, or a PCA if not). Since a random forest approach was used, more details are required about the number of trees and tree size, the division of the data in training, validation, and test subsets and any resulting tuned hyperparameters, and finally whether stated errors refer to out of bag (classification) or misfit (regression) errors. My personal experience is that the default statistical software settings for random forests can lead to non-robust results, thus care must be taken in the selection (and description) of these characteristics.

The metagenomes were analyzed using a random forest regression with permutation test. The choice of a non-parametric, machine-learning approach was due to its higher performance compared to traditional multivariate statistics in the analysis of high-dimensional data (Thompson et al., 2019; Li, 2015; Liu et al., 2019). Random forests are less sensitive to training set size, and are as good at predicting feature importance as neural networks and traditional indicator species approaches. In our study, the random forest was grown using 1000 trees and 4 tree splitting variables. An examination of the mean squared error plot of the random forest indicates that enough trees have been used (Figures 1—figure supplements 2 and 3). The permutation test p-values are obtained using 100 permutation replicates (subsection “in Abundance of carbon metabolism genes across algae cover gradient”, second paragraph; subsection “Central carbon metabolism pathways”, second paragraph). ANOVA and PCA are not appropriate tests in our case, as we are not comparing groups or performing clustering.

d) Since the metagenomes are already assembled the authors should know which species/strain level bacteria are present. Thus I would think a more convincing assessment of the effect of copy number variability would be to look at orthologs and paralogs in at the species/strain level and then evaluate if certain extremes of algal cover (very low, very high) are associated with particular species with many gene replicates or deletions that drive the overall trend. Doing this at the genera level as is the current choice could mute any problematic signal, unless the genera is monospecific. The manuscript states, however, that there is in fact high strain level variation, meaning that this choice is potentially masking a problem. At a minimum, a good reference should justify the choice. And this analysis still does not address the potential complication of variability in gene expression between sites. Finally, from Supplementary file 1 it appears that autotrophic Prochlorococcus has a stronger relationship with algal cover than the heterotrophic bacteria that are the focus of this work. It may be necessary to confirm Prochlorococcus (and Synechococcus, etc.) are not obfuscating your signal.

The taxonomic profiles of the community are reported in Figure 1—source data 1 and discussed (subsection “Abundance of carbon metabolism genes across algal cover gradient”, last paragraph; subsection “Bacterial overflow metabolism in algae-dominated reef”, last paragraph and subsection “Overflow metabolism predicts a shift in ecosystem biomass allocation”, last paragraph). We also included it as Figure 1—source data 1 in the current version. None of these taxa varying with algae cover (including *Prochlorococcus*) display consistent trends in the absence, presence, or copy number variation in the genes with highest importance in the functional analysis. The predictions from the functional metagenomic data from in situ microbial communities were consistent with physiological analysis in vitro, despite the high dimensionality of the data and complexity of environmental influences on microbes

2) Oxygen ebullition experiments a) Based on the supporting data Figure 4—source data 1, it appears that there was a failure to do QA/QC on the oxygen concentrations in the POP experiments. The dissolved oxygen concentration at the end of all of the non-control experiments are outside the stated accurate range for the instrument used, 0-20 mg/L or 0-200% saturation, whichever is lower (https://www.hach.com/hq40d-portable-multi-meter-ph-conductivity-salinity-tds-dissolved-oxygen-do-orp-and-ise-for-water/product-details?id=7640501639). Since this is an optode based optical system, the concentration based range is more notion than reality and the saturation state should always be the limiting bound at high oxygen conditions. Optode based dissolved oxygen sensors have nonlinear responses to concentration, so the deviation from the true value may vary in a way that is not readily predictable (even within the 0-200% range the accuracy varies substantially). Not only are all the experiments beyond the accurate range, many of them (including all the CCA experiments) are beyond the instrument bounds (the internal QA filter is apparently set to 30 mg/L given the constant endpoint values of ~935 μm in many experiments).

As described above, the reviewer mistook the values presented in Figure 4—source data 1. These values are not concentrations (µM), but number of mols in the chamber, accounting for the volume of the incubation chamber, 1.5 L. We used calibrated probed, and performed an uncertainty quantification now included in the Source datasets provided with the manuscript. In the cases where dissolved O_2_ was above the 20 mg/L upper limit, we conservatively used the value 20 mg/L. To test whether ebullition is environmentally relevant we performed additional open-tank experiments (Figure 5—source data 1).

b) Hypothetically the above experimental data from below the upper instrument detection bound could be salvaged, if a full range calibration to Winkler titrations or another sensor were to be run on the same sensor cap originally used, at each of the experimental temperatures (the oxygen sensitivity of the optode foil is temperature dependent). See Bushinsky and Emerson, 2013 for an example of what this might look like (https://doi.org/10.1016/j.marchem.2013.05.001). But the sensor characteristics have a substantial "grow-in" if they are new, and change if they are dry for an extended period, so this would have the best results if the sensor cap had already seen several months of use prior to the experiments and has been kept continuously moist since. But without the beyond detection samples there is insufficient experimental power to determine significance…

The misinterpretation of concentration versus absolute values in Figure 4—source data 1 were now clarified. The uncertainty propagation calculates are provided in source data for Figures 2, 3 and 4, and described in the Materials and methods (–subsection “Uncertainty quantification”).

c) I'm concerned with how the sampling of the headspace gas is described. The manuscript says the gas was transferred from a syringe to a wide mouthed container that the second probe was placed into. Gas diffuses quickly, so I don't see how this could lead to accurate measurements-but I may be misunderstanding, and this refers to the dissolved sample? At any rate this description needs to be clarified.

Air-tight containers and valves were used, and the oxygen concentration in the sample was measured immediately upon collection by transferring the gas volume to the wide-mouth container located next to the incubation chamber. All of the containers and incubation chambers were underwater inside a larger holding tank, and no gas was lost during quantification procedures (subsection “O_2_ release by benthic primary producers”, first paragraph).

d) Setting aside the accuracy issues above, all of the headspace values are approximately in equilibrium with the dissolved oxygen you measured. Henry's law solubility coefficients predict concentrations of 0.36-0.94 mg/mL and 1.3-20.6 mg of oxygen in the gas phase above the water samples-close to the 0.4-0.85 mg/mL and 1.1-17 mg O_2_ measured (assuming 1 atm pressure, and neglecting any lag between production and diffusion). Thus I think that the presence of a headspace isn't proof of the importance of ebullition in a natural environment-it is proof that a highly supersaturated solution will push gas into a headspace (like CO_2_ is in a non-experimental pop bottle). I do suspect bubble nucleation was essential, since the controls were also supersaturated and formed no headspace, but they were a lot less supersaturated so this would be more convincing with evidence of rising bubbles in the chambers-otherwise thermodynamics alone is sufficient without any biophysical interactions. At any rate what the above suggests to me is these chambers are unlikely to reflect environmental outcomes regarding the amount of oxygen released in bubbles. In this light it is very hard to support the statement that in algal dominated reefs 1/3 of the oxygen escapes as bubbles.

We now provide data showing the bubbles being formed on the algae and rising to the surface in an open system, where the oxygen concentration in the water can equilibrate with the atmosphere (Figure 5—source data 1, Videos 1 and 2). No relationship was observed between dissolved oxygen concentrations in the water and ebullition rates, suggesting heterogenous nucleation due to supersaturation on the algae surface microenvironment (Figure 5B, Videos 1 and 2 and subsection “Oxygen production, consumption, and ebullition”, second paragraph).

e) There is no uncertainty analysis of these rates. Given the relatively coarse measurements of bubble volume (to nearest 0.5 mL), short durations (2-4 hours), and uncertainties in saturation state (even when the instrument is within its specified range), both the gas production and dissolved oxygen change must have significant uncertainties, but these are not quantified. I think they should be, and relative uncertainty of each experiment shown so a reader can gauge how significant differences between the box and whisker categories actually are. Also, 2 pts don't make for a convincing rate.

We performed an uncertainty quantification of all incubation data, which is provided in the source data files for Figures 2, 3 and 4. All error propagated from instrument uncertainty was at least five times smaller than statistical standard deviations, and was not considered in the statistical tests (–subsection “Uncertainty quantification”).

3) DOC and oxygen uptake experimentsThe DOC and oxygen uptake ratios in these experiments are far from 1:1, but not in the manner described to support arguments about less complete oxidation of organic matter. To use respiratory quotients with DOC it must be assumed that DOC uptake is equivalent to CO_2_ (DIC) release. The arguments put forward in this manuscript about the incomplete oxidation of organic matter would argue against this for the macroalgae, but I'll set that aside for the moment. At any rate, the -DOC/-O_2_ ratio for the fleshy algae exudate is ~0.9, right in line with theoretical and measured values of +DIC/-O_2_ for bacterioplankton (Robinson 2019, https://doi.org/10.3389/fmars.2018.00533). It is instead the controls and corals that seem to have unexpected values on the order of 0.3. So I see three options: 1) the -DOC/-O_2_ rate ratio shouldn't actually be ~1:1 (or rather 0.9), and thus the discussion of respiratory quotients is a red herring. 2) the corals and controls are doing something unusual that requires an explanation. 3) There are major biases in the rates. I think the error analysis I recommended for the O_2_ production fluxes should be done here as well and could inform this question. So overall I agree that the incubations show more oxygen consumption per unit of DOC consumed with the control and coral exudates than with the algal exudates, but since none of these categories meet the stated expectations this needs to be explored in more detail to confirm that the result is real and robust.

As described above, we disagree with the reviewer statement that “to use respiratory quotients with DOC it must be assumed that DOC uptake is equivalent to CO_2_ (DIC) release”, because a fraction of the DOC consumed by a microbial cell is not oxidized to DIC, but instead allocated into biomass (reviewed in Russel and Cook, 1995). When respiratory quotients were mentioned in the previous version, they referred to primary producer metabolism, not bacterial consumption. We rephrased this section to avoid any confusion (subsection “Ebullition causes O_2_ loss from coral reef”).

The interpretation that an increasing part of the organic carbon consumed by bacteria is not fully oxidized is supported by the decoupling between organic carbon and oxygen consumption (Figure 2), and the higher biomass accumulated per cell (Figure 3).

4) Bacterial community structureThe bacterial community structure is not discussed. How it diverges across sampled environments is challenging to ascertain in Figure 1. I don't find any information on what the bacterial community looked like within the various incubation experiments.

The bacterial community composition results are discussed in subsection “Bacterial overflow metabolism in algae-dominated reef” and subsection “Overflow metabolism predicts a shift in ecosystem biomass allocation”. We also include the data as source data for Figure 1 in this version of the manuscript.

Thus I can't evaluate whether the incubation results are a function of, e.g., incubating a community adapted to one particular environment in a way that would bias the experimental outcomes. While there is a mention about changing community structure with regards to the incubations (Discussion), there is not data in the supplement regarding the community in these experiments, nor would I expect changing genetic composition over the course of experiments lasting hours to days (again, a question for transcriptomics?). This makes it hard to interpret what is happening in the experiments where change in biomass is evaluated. Are changes in biomass due primarily to a single clade? Are certain larger types of bacteria being selected for, or are particular bacteria growing larger while others do not? Is this an expected result based on where those bacteria normally are environmentally successful?

The bacterial inoculum for the experiments was a forereef water column community, which represents the offshore community not yet adapted to a reef environment (subsection “Bacterial DOC and oxygen demands”). These communities reach the reef with incoming currents, and are exposed to reef carbon sources and biogeochemical environment. They are selected in the time frame of water residence times as the offshore water flows in and out of reef lagoons (hours), the same time frame of incubation experiments performed here (Nelson et al., 2011, 2013). In the experiments, bacterial communities are subjected to similar carbon sources and oxygen environment that the offshore communities encounter as they reach coral-dominated or algae-dominated reefs. Therefore, the selection pressures are similar and should lead to the similar physiological responses and genomic adaptation.

Specifically, the changes in cell size observed in situ cannot be attributed to changes in abundance of a single clade. The only clade showing high relative abundance that changed with algae cover was *Prochlorococcus*, which cell volumes comparable to that observed in offshore water, ~ 0.1µm^3^ (Kirchman, 2016), which is the average cell volume observed in the inoculum community of our incubation experiments. The other groups with significant relationships with fleshy algae cover contributed to only 0.9 to 1.4% of the community, and their average cell size is not consistent with the observed increase in cell size: *Thermotoga* has large cells, but their abundance decreased with algae. *Leuconostoc, Rhodopirellula,* and *Methanobacterium* also decrease in abundance with increasing algae. Therefore, none of these clades can explain the change in size observed in this study. We have included this detailed description in the Discussion (subsection “Overflow metabolism predicts a shift in ecosystem biomass allocation”).

Reviewer #2:This study provides a theoretical and experimental based evidence approach to explain the observations for O_2_ depletion in algal dominated coral reefs. The processes of ebullition and microbial metabolism are postulated to drive this lower observed oxygen level in algal dominated reefs. Overall the manuscript is very interesting and details a nice combination of environmental genomics based analysis linked with controlled experimental trials. The postulated physical and biological reasons for reduced O_2_ levels are supported by the experimental data, though the conclusions are still a little speculative. However this approach represents a great approach to hypothesis driven science; and while not able to definitely confirm these process (i.e. ebullition and microbial metabolism) as driving O_2_ depletion, it sets up the concepts to be further tested in environmental settings. Hence the manuscript is worthy of publication with a few comments detailed below highlighting potential areas for improvement.

We thank the reviewer for the comments, which we fully took into consideration to write the revised version, including rephrasing the conclusions to avoid inappropriate extrapolations of the in vitro data to in situ dynamics.

[Editors’ note: the author responses to the re-review follow.]

Thank you for choosing to send your work entitled "Biophysical and physiological processes causing oxygen loss from coral reefs" for consideration at eLife.We thank you for sharing your rebuttal letter. We would welcome a revision of your work for a new cycle of review and note that the revision should address the following concerns:The text is missing or unclear on the rationale for a number of the statistical and analytical choices, and has weaknesses in some of the lines of evidence, including unresolved problems with the oxygen production experiments. Thus the current manuscript, in the opinion of two reviewers, presents the interpretations with a certainty and forcefulness which we did not think is warranted by the present constellation of suggestive (but not definitive) evidence. For an independent opinion, you may consider consulting with Dr. Clara Fuchsman (Horn Point Laboratory, cfuchsman@umces.edu).

We thank the editor and reviewers for the comments. We expanded the Materials and methods to describe the random forests in more detail (subsection “Central carbon metabolism pathways”, second paragraph), and also provided rationale for choosing the random forests (subsection “Abundance of carbon metabolism genes across algal cover gradient”, second paragraph) and other statistical tests performed (–subsection “Statistical tests”).

Regarding the evidences for the oxygen measurements, two important changes were made:

i) Additional experiments were performed to quantify the coral and algae ebullition rates in an open system (Figure 5). These experiments show that ebullition occurs due to bubbles forming on the algae surface. These bubbles detach from the algae and rise though the water column (Videos 1 and 2). There was no relationship between the rates of ebullition and dissolved oxygen concentration in the water column (Figure 5), suggesting that ebullition does not occur due to supersaturation in the water column, but through of heterogenous nucleation on the algae surface. The rates observed in the new experiments are within the range of rates estimated to be necessary to explain the gas produced in the previous POP experiments (Figure 4). A description of how these predicted rates were estimated is provided in the Materials and methods (subsection “Estimates of oxygen loss by ebullition”), and the Results section was updated pointing out that supersaturation was observed in some POP chambers and this issue was addressed in the open-tank experiment (subsection “Oxygen production, consumption, and ebullition”, second paragraph).

ii) Our main conclusion is that microbial respiration and ebullition contribute to the loss of a significant fraction of the oxygen produced by reef primary producers. This conclusion is supported by in vitro experiments in this study. In the current version, we included a weighted linear model (subsection “Predictive model”, Figure 5D) that incorporates the respiration rates from the dark incubation experiments (Figure 2), microbial abundances sustained by primary producers (Figure 3), and oxygen production rates (Figure 4) into an idealized reef with varying coral and algae cover. The model predicted the proportion of the gross oxygen production consumed by microbial respiration and ebullition across the algae cover gradient. This prediction sets a baseline for future in situ studies.

Thank you for clarifying the oxygen units. However, all of the CCA and most of the fleshy macroalgae experiments were beyond the detection bounds of your instrument (20 mg/L), as evidenced by the constant endpoint concentrations (937.5 umol/1.5L=625µM=20 mg/L) across several nominally independent experiments. The headspace concentrations also still appear to reflect equilibrium conditions with the underlying water when the solution is strongly supersaturated (as it is in most of the non-control experiments). How relevant are these experiments for explaining oxygen loss under realistic environmental conditions (less supersaturated and open to the atmosphere)?

We described this issue in the current version (subsection “Oxygen production, consumption, and ebullition”) and performed new open-tank experiments that are closer to realistic environmental conditions due to the equilibration with atmospheric oxygen. These experiments showed significantly high rates of ebullition in algae compared to corals (Figure 5—source data 1, Videos 1 and 2). The ebullition rates were not correlated with the dissolved oxygen concentration in the water, indicating that ebullition occurs due to heterogenous nucleation on algae surfaces. Ebullition has been previously detected in the field on coral reefs (Odum and Odum, 1955, Bellamy and Risk, 1982; Clavier et al., 2008 Freeman, 2018), although its contribution to the oxygen budget has never been quantified. Recently, ebullition was demonstrated to account for the loss of up to 21 and 37% of the O_2_ production in a salt marsh and a lake, respectively (Howard et al., 2018; Koschorreck et al., 2017). This phenomenon may be increasingly important due to the current algae phase-shifts occurring on coral reefs (Figure 5D).

There are statistical techniques for dealing with non-normally distributed data (or more accurately, non-conditionally normal error distributions-A random forest regression may well be a good choice for this analysis, but a rationale should be explained, and the particulars of how it is used clearly stated.

We agree and included a justification for the use of random forest (subsection “Abundance of carbon metabolism genes across algal cover gradient”, second paragraph) and a description of the test and parameters used (subsection “Central carbon metabolism pathways”, second paragraph).

Whether 5 or 16 genes, the correlations to environmental variables need more context. Our concern remains about the effect size versus a null hypothesis of some other random set of genes from your already sequenced genomes, which are far larger than 16 genes. The details required to assess what differences are "real" are currently insufficient. e.g. was the data normalized to bacterial abundance? A single 100mb genome will have far more reads than a 1mb genome, even if there are 10 times more of the smaller-genome organism. You may well have made considered choices for these and other potential issues in your analysis, but you tell the reader none of this, even in the supplement.

We approached the issue of finding genes that are truly correlated with algae cover in two ways:

i) We performed a permutation version of the random forest test, which has an implicit null hypothesis test. The five genes reported were the only ones with a p-value of less than 0.05 in this test (–subsection “Abundance of carbon metabolism genes across algal cover gradient”, second paragraph and legend of Figure 1). The data is not normalized by bacterial abundance because the goal is to test the *relative* enrichment or depletion of genes within a community. Normalizing by bacterial abundance would skew this analysis because an increase in cell abundance would drive the increase in gene abundances even with no relative enrichment for specific functions. We also used control genes, such as rpoB, a RNA polymerase gene that is housekeeping and single-copy, not expected to change in relative abundance across the environmental gradients. Oxidative stress genes did not change in abundance either.

ii) To test the effect of differences in genome sizes and the likelihood of organisms with larger genomes contributing disproportionately to the observed trends, we analyzed the effect of taxonomic composition on the functional changes. The current version includes analyses at genus and species level. The abundance of each taxon was calculated using the FRAP method, which accounts for the probability of a given read mapping to a genome accounting for the genome size (Cobian-Guemes et al., 2016) (–subsection “Central carbon metabolism pathways”, last paragraph). We then searched if the taxa varying in abundance consistently display copy number variations in the genes that significantly vary with algae cover. None of the taxonomic transitions were able to explain the functional trends observed (–subsection “Abundance of carbon metabolism genes across algal cover gradient”, last paragraph). More details on the influence of *Prochlorococcus* abundance are below in the response to reviewers and in the text (subsection “Abundance of carbon metabolism genes across algal cover gradient”, last paragraph; subsection “Bacterial overflow metabolism in algae-dominated reef”, last paragraph, –subsection “Overflow metabolism predicts a shift in ecosystem biomass allocation”, last paragraph).

Regarding your choice of approach for the metagenomic data. Please provide context and defend the choice (and aspects of the analysis of this data) in the text. It would be a good choice to highlight the rationale of this approach in as succinct a manner in the text as you do in your rebuttal letter.

We included the rationale for this analysis at the beginning of the Results section, immediately before we present the metagenomic results: “Different carbon consumption pathways employed by bacteria are associated with varying levels of oxygen consumption rates (Russell and Cook, 1995). […] On coral reefs, metagenomic data reflects the strain-level genomic adaptation that occurs within the timescale of residence time as offshore microbial communities are transported onto the reef environment, where water masses get enriched with benthic exudates changing their biogeochemistry (Nelson et al., 2011, Kelly et al., 2015).”